# THE MULTIBERTS: BERT REPRODUCTIONS FOR ROBUSTNESS ANALYSIS

**Thibault Sellam,**[*] **Steve Yadlowsky,**[*] **Ian Tenney,**[*] **Jason Wei,**[†] **Naomi Saphra,**[‡]
**Alexander D'Amour, Tal Linzen, Jasmijn Bastings, Iulia Turc,**
**Jacob Eisenstein, Dipanjan Das, and Ellie Pavlick**

{`tsellam, yadlowsky, iftenney, epavlick`}@google.com
Google Research

## ABSTRACT

Experiments with pre-trained models such as BERT are often based on a single checkpoint. While the conclusions drawn apply to the *artifact* tested in the experiment (i.e., the particular instance of the model), it is not always clear whether they hold for the more general *procedure* which includes the architecture, training data, initialization scheme, and loss function. Recent work has shown that repeating the pre-training process can lead to substantially different performance, suggesting that an alternate strategy is needed to make principled statements about procedures. To enable researchers to draw more robust conclusions, we introduce the MultiBERTs, a set of 25 BERT-Base checkpoints, trained with similar hyper-parameters as the original BERT model but differing in random weight initialization and shuffling of training data. We also define the Multi-Bootstrap, a non-parametric bootstrap method for statistical inference designed for settings where there are multiple pre-trained models and limited test data. To illustrate our approach, we present a case study of gender bias in coreference resolution, in which the Multi-Bootstrap lets us measure effects that may not be detected with a single checkpoint. We release our models and statistical library,[1] along with an additional set of 140 intermediate checkpoints captured during pre-training to facilitate research on learning dynamics.

## 1 INTRODUCTION

Contemporary natural language processing (NLP) relies heavily on pretrained language models, which are trained using large-scale unlabeled data (Bommasani et al., 2021). BERT (Devlin et al., 2019) is a particularly popular choice: it has been widely adopted in academia and industry, and aspects of its performance have been reported on in thousands of research papers (see, e.g., Rogers et al., 2020, for an overview). Because pre-training large language models is computationally expensive (Strubell et al., 2019), researchers often rely on the release of model checkpoints through libraries such as HuggingFace Transformers (Wolf et al., 2020), which enable them to use large-scale language models without repeating the pre-training work. Consequently, most published results are based on a small number of publicly released model checkpoints.

While this reuse of model checkpoints has lowered the cost of research and facilitated head-to-head comparisons, it limits our ability to draw general scientific conclusions about the performance of a particular class of models (Dror et al., 2019; D'Amour et al., 2020; Zhong et al., 2021). The key issue is that reusing model checkpoints makes it hard to generalize observations about the behavior of a single model *artifact* to statements about the underlying pre-training *procedure* which created it. Pre-training such models is an inherently stochastic process which depends on the initialization of the model's parameters and the ordering of training examples; for example, D'Amour et al.

---

[*] Equal contribution.
[†] Work done as a Google AI resident.
[‡] Work done during an internship at Google.
[1] `http://goo.gle/multiberts`

(2020) report substantial quantitative differences across multiple checkpoints of the same model architecture on several "stress tests" (Naik et al., 2018; McCoy et al., 2019). It is therefore difficult to know how much of the success of a model based on the original BERT checkpoint is due to BERT's design, and how much is due to idiosyncrasies of a particular artifact. Understanding this difference is critical if we are to generate reusable insights about deep learning for NLP, and improve the state-of-the-art going forward (Zhou et al., 2020; Dodge et al., 2020; Aribandi et al., 2021).

This paper describes the MultiBERTs, an effort to facilitate more robust research on the BERT model. Our primary contributions are:

- We release the MultiBERTs, a set of 25 BERT-Base, Uncased checkpoints to facilitate studies of robustness to parameter initialization and order of training examples (§2). Releasing these models preserves the benefits to the community of a single checkpoint release (i.e., low cost of experiments, apples-to-apples comparisons between studies based on these checkpoints), while enabling researchers to draw more general conclusions about the BERT pre-training procedure.

- We present the Multi-Bootstrap, a non-parametric method to quantify the uncertainty of experimental results based on multiple pre-training seeds (§3), and provide recommendations for how to use the Multi-Bootstrap and MultiBERTs in typical experimental scenarios. We implement these recommendations in a software library.

- We illustrate the approach with a practical use case: we investigate the impact of counterfactual data augmentation on gender bias, in a BERT-based coreference resolution systems (Webster et al., 2020) (§4). Additional examples are provided in Appendix E, where we document challenges with reproducing the widely-used original BERT checkpoint.

The release also includes an additional 140 intermediate checkpoints, captured during training for 5 of the runs (28 checkpoints per run), to facilitate studies of learning dynamics. Our checkpoints and statistical libraries are available at: `http://goo.gle/multiberts`.

**Additional Related Work.** The MultiBERTs release builds on top of a large body of work that seeks to analyze the behavior of BERT (Rogers et al., 2020). In addition to the studies of robustness cited above, several authors have introduced methods to reduce BERT's variability during fine-tuning (Zhang et al., 2021; Mosbach et al., 2021; Dodge et al., 2020; Lee et al., 2020; Phang et al., 2018). Other authors have also studied the time dimension, which motivates our release of intermediate checkpoints (Liu et al., 2021; Hao et al., 2020; Saphra & Lopez, 2019; Chiang et al., 2020; Dodge et al., 2020). Similarly to §3, authors in the NLP literature have recommended best practices for statistical testing (Koehn, 2004; Dror et al., 2018; Berg-Kirkpatrick et al., 2012; Card et al., 2020; Søgaard et al., 2014; Peyrard et al., 2021), many of which are based on existing tests to estimate the uncertainty of test sample. In concurrent work, Deutsch et al. (2021) considered bootstrapping methods similar to the Multi-Bootstrap, in the context of summarization metrics evaluation. Also in concurrent work, the Mistral project (Karamcheti et al., 2021) released a set of 10 GPT-2 models with intermediate checkpoints at different stages of pre-training. Our work is complementary, focusing on BERT, introducing a larger number of pre-training seeds, and presenting a methodology to draw robust conclusions about model performance.

## 2  RELEASE DESCRIPTION

We first describe the MultiBERTs release: how the checkpoints were trained and how their performance compares to the original BERT on two common language understanding benchmarks.

### 2.1  TRAINING

**Overview.** The MultiBERTs checkpoints are trained following the code and procedure of Devlin et al. (2019), with minor hyperparameter modifications necessary to obtain comparable results on GLUE (Wang et al., 2019); a detailed discussion of these differences is provided in Appendix E. We use the BERT-Base, Uncased architecture with 12 layers and embedding size 768. We trained the models on a combination of BooksCorpus (Zhu et al., 2015) and English Wikipedia. Since the

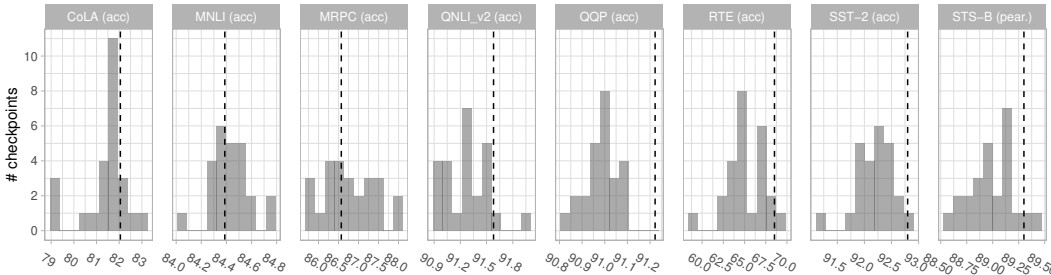

Figure 1: Distribution of the performance on GLUE dev sets (Wang et al., 2019), averaged across fine-tuning runs for each checkpoint. The dashed line indicates the performance of the original BERT release.

exact dataset used to train the original BERT is not available, we used a more recent version that was collected by Turc et al. (2019) with the same methodology.

**Checkpoints.** We release 25 models trained for two million steps each, each training step involving a batch of 256 sequences. For five of these models, we release 28 additional checkpoints captured over the course of pre-training (every 20,000 training steps up to 200,000, then every 100,000 steps). In total, we release 165 checkpoints, about 68 GB of data.

**Training Details.** As in the original BERT paper, we used batch size 256 and the Adam optimizer (Kingma & Ba, 2014) with learning rate 1e-4 and 10,000 warm-up steps. We used the default values for all the other parameters, except the number of steps, which we set to two million, and sequence length, which we set to 512 from the beginning with up to 80 masked tokens per sequence.[2] We follow the BERT code and initialize the layer parameters from a truncated normal distribution, using mean 0 and standard deviation 0.02. We train using the same configuration as Devlin et al. (2019)[3], with each run taking about 4.5 days on 16 Cloud TPU v2 chips.

**Environmental Impact Statement.** We estimate compute costs at around 1728 TPU-hours for each pre-training run, and around 208 GPU-hours plus 8 TPU-hours for associated fine-tuning experiments (§2.2, including hyperparameter search and 5x replication). Using the calculations of Luccioni et al. (2019)[4], we estimate this as about 250 kg CO2e for each of our 25 models. Counting the 25 runs each of CDA-incr and CDA-full from §4, associated coreference models (20 GPU-hours per pretraining model), and additional experiments of Appendix E, this gives a total of about 12.0 metric tons CO2e before accounting for offsets or clean energy. Based on the report by Patterson et al. (2021) of 78% carbon-free energy in Google Iowa (us-central1), we estimate that reproducing these experiments would emit closer to 2.6 tons CO2e, or slightly more than two passengers on a round-trip flight between San Francisco and New York. By releasing the trained checkpoints publicly, we aim to enable many research efforts on reproducibility and robustness without requiring this cost to be incurred for every subsequent study.

## 2.2 PERFORMANCE BENCHMARKS

**GLUE Setup.** We report results on the development sets of the GLUE tasks: CoLA (Warstadt et al., 2019), MNLI (matched) (Williams et al., 2018), MRPC (Dolan & Brockett, 2005), QNLI (v2) (Rajpurkar et al., 2016; Wang et al., 2019), QQP (Chen et al., 2018), RTE (Bentivogli et al., 2009), SST-2 (Socher et al., 2013), and SST-B (Cer et al., 2017). In all cases we follow the same approach as Devlin et al. (2019). For each task, we fine-tune BERT for 3 epochs using a batch

---

[2]Specifically, we keep the sequence length constant (the paper uses 128 tokens for 90% of the training then 512 for the remaining 10%) to expose the model to more tokens and simplify the implementation. As we were not able to reproduce original BERT exactly using either 1M or 2M steps (see Appendix E for discussion), we release MultiBERTs trained with 2M steps under the assumption that higher-performing models are more interesting objects of study.

[3]We use https://github.com/google-research/bert with TensorFlow (Abadi et al., 2015) version 2.5 in v1 compatibility mode.

[4]https://mlco2.github.io/impact/

size of 32. We run a parameter sweep on learning rates [5e-5, 4e-5, 3e-5, 2e-5] and report the best score. We run the procedure five times for each of the 25 models and average the results.

**SQuAD Setup.** We report results on the development sets of SQuAD versions 1.1 and 2.0 (Rajpurkar et al., 2016; 2018), using a setup similar to that of Devlin et al. (2019). For both sets of experiments, we use batch size 48, learning rate 5e-5, and train for 2 epochs.

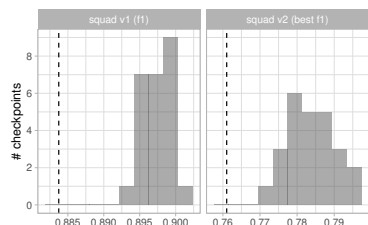

Figure 2: Performance distribution on the dev sets of SQuAD v1.1 and v2.0 (Rajpurkar et al., 2016; 2018).

**Results.** Figures 1 and 2 show the distribution of the MultiBERTs models' performance on the development sets of GLUE and SQuAD, in comparison to the original BERT checkpoint.[5] On most tasks, original BERT's performance falls within the same range as MultiBERTs (i.e., original BERT is between the minimum and maximum of the MultiBERTs' scores). However, original BERT outperforms all MultiBERTs models on QQP, and under-performs them on SQuAD. The discrepancies may be explained by both randomness and differences in training setups, as investigated further in Appendix E.

To further illustrate the performance variability inherent to pre-training and fine-tuning, we analyze the instance-level agreement between the models in Appendix C.

## 3   HYPOTHESIS TESTING USING MULTIPLE CHECKPOINTS

The previous section compared MultiBERTs with the original BERT, finding many similarities but also some differences (e.g., in the case of SQuAD). To what extent can these results be explained by random noise? More generally, how can we quantify the uncertainty of a set of experimental results when there are multiple sources of randomness?

In parallel to the MultiBERTs release, we propose a more principled and standardized method to compare training procedures. We recommend a non-parametric bootstrapping procedure, the "Multi-Bootstrap", which enables us to make inference about model performance in the face of multiple sources of uncertainty: the randomness due to the pre-training seed, the fine-tuning seed, and the finite test data. The main idea is to use the average behavior over seeds as a means of summarizing expected behavior in an ideal world with infinite samples.

Although we present Multi-Bootstrap in the context of analyzing the MultiBERTs, the method could be applied in all setups that involve a set of checkpoints pre-trained with the same method, a finite test set, and (possibly) multiple rounds of fine-tuning. The Multi-Bootstrap is implemented as a Python library, included with the MultiBERTs release.

### 3.1   INTERPRETING STATISTICAL RESULTS

The Multi-Bootstrap provides an estimate of the amount of remaining uncertainty when summarizing the performance over multiple seeds. The following notation will help us state this precisely. We assume access to model predictions $f(x)$ for each instance $x$ in the evaluation set. We consider randomness arising from:

1. The choice of pre-training seed $S \sim M$
2. The choice of fine-tuning seed $T \sim N$
3. The choice of test sample $(X, Y) \sim D$

The Multi-Bootstrap procedure allows us to account for all of the above. Specifically, MultiBERTs enables us to estimate the variance due to the choice of pre-training seed (1), which would not be possible with a single artifact. Note that multiple fine-tuning runs are not required in order to use the procedure.

---

[5]We used `https://storage.googleapis.com/bert_models/2020_02_20/uncased_L-12_H-768_A-12.zip`, as linked from `https://github.com/google-research/bert`.

For each pre-training seed $s$, let $f_s(x)$ denote the learned model's prediction on input features $x$ and let $L(s)$ denote the expected performance metric of $f_s$ on a test distribution $D$ over features $X$ and labels $Y$. For example, the accuracy would be $L(s) = \mathbb{E}[1\{Y = f_s(X)\}]$. We can use the test sample (which we will assume has $n_x$ examples) to estimate the performance for each of the seeds in MultiBERTs, which we denote as $\widehat{L}(s)$.

The performance $L(s)$ depends on the seed, but we are interested in summarizing the model over all seeds. A natural summary is the average over seeds, $\mathbb{E}_{S \sim M}[L(S)]$, which we will denote by $\theta$. Then, using $n_s$ independently sampled seeds, we can compute an estimate $\widehat{\theta}$ as

$$\widehat{\theta} = \frac{1}{n_s} \sum_{j=1}^{n_s} \widehat{L}(S_j) \,.$$

Because $\widehat{\theta}$ is computed under a finite evaluation set and finite number of seeds, it is necessary to quantify the uncertainty of the estimate. The goal of Multi-Bootstrap is to estimate the distribution of the error in this estimate, $\widehat{\theta} - \theta$, in order to compute confidence intervals and test hypotheses about $\theta$, such as whether it is above some threshold of interest. Below, we describe a few common experimental designs in NLP that can be studied with these tools.

**Design 1: Comparison to a Fixed Baseline.** In many use cases, we want to compare BERT's behavior to that of a single, fixed baseline. For instance, does BERT encode information about syntax as a feature-engineered model would (Tenney et al., 2019; Hewitt & Manning, 2019)? Does it encode social stereotypes, and how does it compare to human biases (Nadeem et al., 2021)? Does it encode world knowledge, similarly to explicit knowledge bases (Petroni et al., 2019)? Does another model such as RoBERTa (Liu et al., 2019) outperform BERT on common tasks such as those from the GLUE benchmark?

In all these cases, we compare MultiBERTs to some external baseline of which we only have a single estimate (e.g., random or human performance), or against an existing model that is not derived from the MultiBERTs checkpoints. We treat the baseline as fixed, and assess only the uncertainty that arises from MultiBERTs' random seeds and the test examples.

**Design 2: Paired Samples.** Alternatively, we might seek to assess the effectiveness of a specific intervention on model behavior. In such studies, an intervention is proposed (e.g., representation learning via a specific intermediate task, or a specific architecture change) which can be applied to any pre-trained BERT checkpoint. The question is whether the procedure results in an improvement over the original BERT pre-training method: does the intervention reliably produce the desired effect, or is the observed effect due to the idiosyncrasies of a particular model artifact? Examples of such studies include: Does intermediate tuning on NLI after pre-training make models more robust across language understanding tasks (Phang et al., 2018)? Does pruning attention heads degrade model performance on downstream tasks (Voita et al., 2019)? Does augmenting BERT with information about semantic roles improve performance on benchmark tasks (Zhang et al., 2020)?

We refer to studies like the above as *paired* since each instance of the baseline model $f_s$ (which does not receive the intervention) can be paired with an instance of the proposed model $f'_s$ (which receives the stated intervention) such that $f_s$ and $f'_s$ are based on the same pretrained checkpoint produced using the same seed. Denoting $\theta_f$ and $\theta_{f'}$ as the expected performance defined above for the baseline and intervention model respectively, our goal is to test hypotheses about the true difference in performance $\delta = \theta_{f'} - \theta_f$ using the estimated difference $\widehat{\delta} = \widehat{\theta}_{f'} - \widehat{\theta}_f$.

In a paired study, Multi-Bootstrap allows us to estimate both of the errors $\widehat{\theta}_f - \theta_f$ and $\widehat{\theta}_{f'} - \theta_{f'}$, as well as the correlation between the two. Together, these allow us to approximate the distribution of the overall estimation error $\widehat{\delta} - \delta = (\widehat{\theta}_f - \widehat{\theta}_{f'}) - (\theta_f - \theta_{f'})$, between the estimate $\widehat{\delta}$ and the truth $\delta$. With this, we can compute confidence intervals for $\delta$, the true average effect of the intervention on performance over seeds, and test hypotheses about $\delta$, as well.

**Design 3: Unpaired Samples.** Finally, we might seek to compare a number of seeds for both the intervention and baseline models, but may not expect them to be aligned in their dependence on the seed. For example, the second model may use a different architecture so that they cannot be built

from the same checkpoints, or the models may be generated from entirely separate initialization schemes. We refer to such studies as *unpaired*. Like in a paired study, the Multi-Bootstrap allows us to estimate the errors $\widehat{\theta}_f - \theta_f$ and $\widehat{\theta}_{f'} - \theta_{f'}$; however, in an unpaired study, we cannot estimate the correlation between the errors. Thus, we assume that the correlation is zero. This will give a conservative estimate of the error $(\widehat{\theta}_f - \widehat{\theta}_{f'}) - (\theta_f - \theta_{f'})$, as long as $\widehat{\theta}_f - \theta_f$ and $\widehat{\theta}_{f'} - \theta_{f'}$ are not negatively correlated. Since there is little reason to believe that the random seeds used for two different models would induce a negative correlation between the models' performance, we take this assumption to be relatively safe.

**Hypothesis Testing.** Given the measured uncertainty, we recommend testing whether or not the difference is meaningfully different from some arbitrary predefined threshold (i.e., $0$ in the typical case). Specifically, we are often interested in rejecting the null hypothesis that the intervention does not improve over the baseline model, i.e.,

$$H_0 : \delta \leq 0 \tag{1}$$

in a statistically rigorous way. This can be done with the Multi-Bootstrap procedure described below.

## 3.2 MULTI-BOOTSTRAP PROCEDURE

The *Multi-Bootstrap* is a non-parametric bootstrapping procedure that allows us to estimate the distribution of the error $\widehat{\theta} - \theta$ over the seeds *and* test instances. The algorithm supports both paired and unpaired study designs, differentiating the two settings only in the way the sampling is performed.

To keep the presentation simple, we will assume that the performance $L(s)$ is an average of a per-example metric $\ell(x, y, f_s)$ over the distribution $D$ of $(X, Y)$, such as accuracy or the log likelihood, and $\widehat{L}(s)$ is similarly an empirical average with the observed $n_x$ test examples,

$$L(s) = \mathbb{E}_D[\ell(X, Y, f_s)], \text{ and } \widehat{L}(s) = \frac{1}{n_x} \sum_{i=1}^{n_x} \ell(X_i, Y_i, f_s).$$

We note that the mapping $D \mapsto L(s)$ is linear in $D$, which is required for our result in Theorem 1. However, we conjecture that this is an artifact of the proof; like most bootstrap methods, the method here likely generalizes to any performance metric which behaves asymptotically like a linear mapping of $D$, including AUC, BLEU score (Papineni et al., 2002), and expected calibration error.

Building on the rich literature on bootstrap methods (e.g., Efron & Tibshirani, 1994), the Multi-Bootstrap is a new procedure which accounts for the way that the combined randomness from the seeds and test set creates error in the estimate $\widehat{\theta}$. The statistical underpinnings of this approach have theoretical and methodological connections to inference procedures for two-sample tests (Van der Vaart, 2000), where the samples from each population are independent. However, in those settings, the test statistics naturally differ as a result of the scientific question at hand.

In our procedure, we generate a bootstrap sample from the full sample with replacement separately over both the randomness from the pre-training seed $s$ and from the test set $(X, Y)$. That is, we generate a sample of pre-training seeds $(S_1^*, S_2^*, \ldots, S_{n_s}^*)$ with each $S_j^*$ drawn randomly with replacement from the pre-training seeds, and we generate a test set sample $((X_1^*, Y_1^*), (X_2^*, Y_2^*), \ldots, (X_{n_x}^*, Y_{n_x}^*))$ with each $(X, Y)$ pair drawn randomly with replacement from the full test set. Then, we compute the bootstrap estimate $\widehat{\theta}^*$ as

$$\widehat{\theta}^* = \frac{1}{n_s} \sum_{j=1}^{n_s} \widehat{L}^*(S_j^*), \text{ where } \widehat{L}^*(s) = \frac{1}{n_x} \sum_{i=1}^{n_x} \ell(X_i^*, Y_i^*, f_s).$$

To illustrate the procedure, we present a minimal Python implementation in Appendix A. For sufficiently large $n_x$ and $n_s$, the distribution of the estimation error $\widehat{\theta} - \theta$ is approximated well by the distribution of $\widehat{\theta}^* - \widehat{\theta}$ over re-draws of the bootstrap samples, as stated precisely in Theorem 1.

**Theorem 1.** *Assume that* $\mathbb{E}[\ell^2(X, Y, f_S)] < \infty$. *Furthermore, assume that for each $s$,* $\mathbb{E}[\ell^2(X, Y, f_s)] < \infty$, *and for almost every $(x, y)$ pair,* $\mathbb{E}[\ell^2(X, Y, f_S) \mid X = x, Y = y] < \infty$. *Let $n = n_x + n_s$, and assume that $0 < p_s = n_s/n < 1$ stays fixed (up to rounding error) as $n \to \infty$. Then, there exists $0 < \sigma^2 < \infty$ such that $\sqrt{n}(\widehat{\theta} - \theta) \xrightarrow{d} G$ with $G \sim \mathcal{N}(0, \sigma^2)$. Furthermore, conditionally on $((X_1, Y_1), (X_2, Y_2), \ldots)$, $\sqrt{n}(\widehat{\theta}^* - \widehat{\theta}) \xrightarrow{d} G$.*

The proof of Theorem 1 is in Appendix B, along with a comment on the rate of convergence for the approximation error. The challenge with applying existing theory to our method is that while the seeds and data points are each marginally iid, the observed losses depend on both, and therefore are not iid. Therefore, we need to handle this non-iid structure in our method and proof.

For nested sources of randomness (e.g., if for each pre-training seed $s$, we have estimates from multiple fine-tuning seeds), we average over all of the inner samples (fine-tuning seeds) in every bootstrap sample, motivated by Field & Welsh (2007)'s recommendations for bootstrapping clustered data.

**Paired Samples (design 2, continued).** In a paired design, the Multi-Bootstrap procedure can additionally tell us the joint distribution of $\widehat{\theta}_{f'} - \theta_{f'}$ and $\widehat{\theta}_f - \theta_f$. To do so, one must use the same bootstrap samples of the seeds $(S_1^*, S_2^*, \ldots, S_{n_s}^*)$ and test examples $((X_1^*, Y_1^*), (X_2^*, Y_2^*), \ldots, (X_{n_x}^*, Y_{n_x}^*))$ for both models. Then, the correlation between the errors $\widehat{\theta}_{f'} - \theta_{f'}$ and $\widehat{\theta}_f - \theta_f$ is well approximated by the correlation between the bootstrap errors $\widehat{\theta}_{f'}^* - \theta_{f'}^*$ and $\widehat{\theta}_f^* - \theta_f^*$.

In particular, recall that we defined the difference in performance between the intervention $f'$ and the baseline $f$ to be $\delta$, and defined its estimator to be $\widehat{\delta}$. With the Multi-Bootstrap, we can estimate the bootstrapped difference

$$\widehat{\delta}^* = \widehat{\theta}_{f'}^* - \widehat{\theta}_f^*.$$

With this, the distribution of the estimation error $\widehat{\delta} - \delta$ is well approximated by the distribution of $\widehat{\delta}^* - \widehat{\delta}$ over bootstrap samples.

**Unpaired Samples (design 3, continued).** For studies that do not match the paired format, we adapt the Multi-Bootstrap procedure so that, instead of sampling a single pre-training seed that is shared between $f$ and $f'$, we sample pre-training seeds for each one independently. The remainder of the algorithm proceeds as in the paired case. Relative to the paired design discussed above, this additionally assumes that the errors due to differences in pre-training seed between $\widehat{\theta}_{f'} - \theta_{f'}$ and $\widehat{\theta}_f - \theta_f$ are independent.

**Comparison to a Fixed Baseline (design 1, continued).** Often, we do not have access to multiple estimates of $L(s)$, for example, when the baseline $f$ against which we are comparing is an estimate of human performance for which only mean accuracy was reported, or when $f$ is the performance of a previously-published model for which there only exists a single artifact or for which we do not have direct access to model predictions. When we have only a point estimate $\widehat{\theta}_f = \widehat{L}(S_1)$ of $\theta_f$ for the baseline $f$ with a single seed $S_1$, we recommend using Multi-Bootstrap to compute a confidence interval around $\theta_{f'}$ and reporting where the given estimate of baseline performance falls within that distribution. An example of such a case is Figure 1, in which the distribution of MultiBERTs performance is compared to that from the single checkpoint of the original BERT release. In general such results should be interpreted conservatively, as we cannot make any claims about the variance of the baseline model.

**Hypothesis Testing.** A valid $p$-value for the hypothesis test described in Equation 1 is the fraction of bootstrap samples from the above procedure for which the estimate $\widehat{\delta}$ is negative.

## 4 APPLICATION: GENDER BIAS IN COREFERENCE SYSTEMS

We present a case study to illustrate how MultiBERTs and the Multi-Bootstrap can help us draw more robust conclusions about model behavior.

*The use case is based on gendered correlations. For a particular measure of gender bias, we take a single BERT checkpoint and measure a value of 0.35. We then apply an intervention, foo, designed to reduce this correlation, and measure 0.25. In an effort to do even better, we create a whole new checkpoint by applying the foo procedure from the very beginning of pre-training. On this checkpoint, we measure 0.3. How does one make sense of this result?*

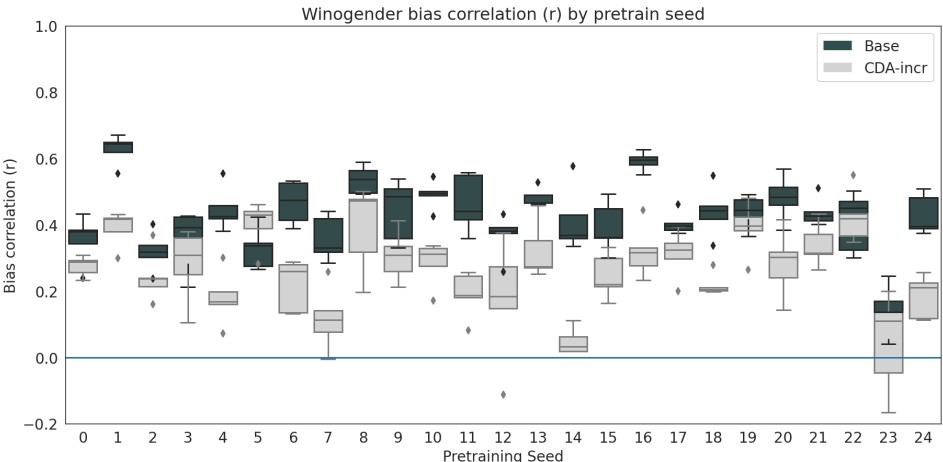

Figure 3: Bias correlation on Winogender for each pre-training seed. Each box represents the distribution of the score over five training runs of the coreference model. Dark boxes represent each base MultiBERTs checkpoint, while lighter boxes (CDA-incr) are the corresponding checkpoints after 50k steps of additional pretraining with CDA. Some seeds are better than others on this task (for example, seed 23), but the CDA-incr consistently reduces the bias correlation for most seeds.

As a concrete example, we analyze gender bias in coreference systems (Rudinger et al., 2018) and showing how MultiBERTs and the Multi-Bootstrap can help us understand the effect of an intervention, counterfactual data augmentation (CDA). We follow a set-up similar to Webster et al. (2020), which augments the BERT pretraining data with counterfactual sentences created by randomly swapping English binary-gendered pronouns. The goal is to weaken the correlation between gendered pronouns and other words such as occupation terms (e.g., doctor, nurse). We compare our baseline MultiBERTs models to two strategies for CDA. In the first (**CDA-incr**), we continue pre-training each MultiBERTs model for an additional 50K steps on the counterfactual data of Webster et al. (2020). In the second, we train BERT models from scratch (**CDA-full**) on the same dataset.

The Winogender dataset consists of template sentences covering 60 occupation terms and instantiated with either male, female, or neutral pronouns. We follow Webster et al. (2020) and train a gold-mention coreference system using a two-layer feedforward network that takes span representations from a frozen BERT encoder as input and makes binary predictions for mention-referent pairs. The model is trained on OntoNotes (Hovy et al., 2006) and evaluated on the Winogender examples for both per-sentence accuracy and a bias score, defined as the Pearson correlation between the per-occupation bias score (Figure 4 of Rudinger et al. 2018) and the occupational gender statistics from the U.S. Bureau of Labor Statistics.[6] For each pre-training run, we train five coreference models, using the same encoder but different random seeds to initialize the classifier weights and to shuffle the training data.

## 4.1 PAIRED ANALYSIS: CDA-INCR VS. BASE

We investigate the impact of the intervention on performance and bias. Overall accuracy is fairly consistent across pre-training seeds, at 62.6±1.2% for the base model, with only a small and not statistically significant change under CDA-incr (Table 1). However, as shown in Figure 3, there is considerable variation in bias correlation, with $r$ values between 0.1 and 0.7 depending on pre-training seed.[7] The range for CDA-incr overlaps somewhat, with values between 0.0 and 0.4; however, because the incremental CDA is an intervention on each base checkpoint, we can look at the individual seeds and see that in most cases there appears to be a significant improvement. A paired Multi-Bootstrap allows us to quantify this and further account for noise due to the finite evaluation

---

[6]We use the occupation data as distributed with the Winogender dataset, https://github.com/rudinger/winogender-schemas.

[7]Some of this variation is due to the classifier training, but on this task there is a large intrinsic contribution from the pretraining seed. See Appendix D for a detailed analysis.

sample of 60 occupations. The results are shown in Table 1, which show that CDA-incr significantly reduces bias by $\widehat{\delta} = -0.162$ with $p = 0.001$.

|  |  | Accuracy | Bias Corr. ($r$) |
|---|---|---|---|
| Base | $\theta_f$ | 0.626 | 0.423 |
| CDA-incr | $\theta_{f'}$ | 0.623 | 0.261 |
| Avg. Diff. | $\delta = \theta_{f'} - \theta_f$ | -0.004 | -0.162 |
| $p$-value |  | 0.210 | 0.001 |

Table 1: Paired Multi-Bootstrap results for CDA intervention over the base MultiBERTs checkpoints on Winogender. Accuracy is computed by bootstrapping over all 720 examples, while for bias correlation we first compute per-occupation bias scores and then bootstrap over the 60 occupation terms. For both, we use 1,000 bootstrap samples. A lower value of $r$ indicates less gender-occupation bias.

|  |  | Accuracy | Bias Corr. ($r$) | Seeds | Examples |
|---|---|---|---|---|---|
| CDA-incr | $\theta_f$ | 0.623 | 0.256 | 0.264 | 0.259 |
| CDA-full | $\theta_{f'}$ | 0.622 | 0.192 | 0.194 | 0.193 |
| Avg. Diff. | $\delta = \theta_{f'} - \theta_f$ | -0.001 | -0.064 | -0.070 | -0.067 |
| $p$-value |  | 0.416 | 0.132 | 0.005 | 0.053 |

Table 2: Unpaired Multi-Bootstrap results comparing CDA-full to CDA-incr on Winogender. Examples are treated as in Figure 1. The "Seeds" column bootstraps only over pre-training seeds while using the full set of 60 occupations, while the "Examples" column bootstraps over examples, averaging over all pre-training seeds. For all tests we use 1,000 bootstrap samples.

## 4.2 Unpaired analysis: CDA-full vs. CDA-incr

We can also test if we get any additional benefit from running the entire pre-training with counterfactually-augmented data. Similar to MultiBERTs, we trained 25 CDA-full checkpoints for 2M steps on the CDA dataset.[8] Because these are entirely new checkpoints, independent from the base MultiBERTs runs, we use an *unpaired* version of the Multi-Bootstrap, which uses the same set of examples but samples pretraining seeds independently for CDA-incr and CDA-full. As shown in Table 2, overall accuracy does not change appreciably (0.622 vs. 0.623, $p = 0.416$), while bias correlation seems to decrease but not significantly (0.256 vs 0.192, $\delta$ = -0.064 with $p = 0.132$).

As an ablation, we also experiment with sampling over either only seeds (taking the set of examples, i.e. occupations, as fixed), or over examples (taking the set of 25 seeds as fixed). As shown in Table 2, we find lower $p$-values (0.005 and 0.053) in both cases—showing that failing to account for finite samples along *either* dimension could lead to overconfident conclusions.

In Appendix E, we present two additional examples: a paired study where we increase pre-training time from 1M to 2M steps, as well as an unpaired comparison to the original `bert-base-uncased` checkpoint.

## 5 Conclusion

To make progress on language model pre-training, it is essential to distinguish between the properties of specific model artifacts and those of the training procedures that generated them. To this end, we have presented two resources: the MultiBERTs, a set of 25 model checkpoints to support robust research on BERT, and the Multi-Bootstrap, a non-parametric statistical method to estimate the uncertainty of model comparisons across multiple training seeds. We demonstrated the utility of these resources by showing how to quantify the effect of an intervention to reduce a type of gender bias in coreference systems built on BERT. We hope that the release of multiple checkpoints and the use of principled hypothesis testing will become standard practices in research on pre-trained language models.

---

[8]Following Webster et al. (2020), we use 20 masks per sequence instead of the 80 from Devlin et al. (2019).

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

## A    MINIMAL IMPLEMENTATION OF THE MULTI-BOOTSTRAP

Below, we present a simplified Python implementation of the Multi-Bootstrap algorithm presented in Section 3.2. It describes a single-sided version of the procedure, which could be used, e.g., to test that a model's performance is greater than $0$. The input is a matrix of predictions where row indices correspond to test examples and column indices to random seeds. The functions returns an array of $n_{boot}$ samples $[\widehat{\theta}_1, \ldots, \widehat{\theta}_{n_{boot}}]$.

```python
def multibootstrap(predictions, labels, metric_fun, nboot):
  """
  Generates bootstrap samples of a model's performance.

  Input:
    predictions: 2D Numpy array with the predictions for different seeds.
    labels: 1D Numpy array with the labels.
    metric_fun: Python function. Takes a pair of arrays as input, and
    returns a metric or loss.
    nboot: Number of bootstrap samples to generate.

  Output:
    Numpy array with nboot samples.

  """
  # Checks the data format.
  n_samples, n_seeds = predictions.shape
  assert labels.shape == (n_samples,)

  thetas = np.zeros(nboot)
  for boot_ix in range(nboot):
    # Samples n_samples test examples and n_seeds pre-training seeds.
    x_samples = np.random.choice(n_samples, size=n_samples, replace=True)
    s_samples = np.random.choice(n_seeds, size=n_seeds, replace=True)

    # Computes the metric over the bootstrapping samples.
    sampled_predictions = predictions[np.ix_(x_samples, s_samples)]
    sampled_labels = labels[x_samples]
    sampled_metrics = [
      metric_fun(sampled_predictions[:,j], sampled_labels)
      for j in range(n_seeds)
    ]

    # Averages over the random seeds.
    thetas[boot_ix] = np.mean(sampled_metrics)

  return thetas
```

We provide the complete version of the algorithm on our repository http://goo.gle/multiberts. Our implementation is optimized and supports all the experiment designs described in Section 3, including paired and unpaired analysis as well as multiple fine-tuning runs for each pretraining seed.

## B  PROOF OF THEOREM 1

Before giving the proof, we define some useful notation that will simplify the argument considerably. We let $D_n$ be the empirical measure over the $n_x$ observations $(Z_i = (X_i, Y_i))_{i=1}^n$, and $M_n$ be the empirical measure over the $n_s$ observations $(S_j)_{j=1}^n$. For a function $f : V \to \mathbb{R}$ and a distribution $P$ over $V$, we will use the shorthand $Pf$ to denote the expectation of $f$ under $P$,

$$Pf = \mathbb{E}_{V \sim P}[f(V)].$$

For example, this allows us to write

$$\theta = DM\ell = \mathbb{E}_{Z \sim D} \mathbb{E}_{S \sim M} \ell(Z, f_S), \text{ and } \widehat{\theta} = D_n M_n \ell = \frac{1}{n_x} \sum_{i=1}^{n_x} \frac{1}{n_s} \sum_{j=1}^{n_s} \ell(Z_i, f_{S_j}).$$

For the bootstrapped distributions, let $D_n^*$ denote the distribution over the bootstrap data samples $(Z_1^*, Z_2^*, \ldots, Z_{n_x}^*)$ and $M_n^*$ denote the distribution over the bootstrapped seed samples, $(S_1^*, S_2^*, \ldots, S_{n_s}^*)$, both conditional on the observed samples $(Z_i)_{i=1}^{n_x}$ and $(S_j)_{j=1}^{n_s}$. Note that the empirical average over a bootstrapped sample

$$\frac{1}{n_x} \sum_{i=1}^{n_x} \frac{1}{n_s} \sum_{j=1}^{n_s} \ell(Z_i^*, f_{S_j^*})$$

can be written as

$$\frac{1}{n_x} \sum_{i=1}^{n_x} \frac{1}{n_s} \sum_{j=1}^{n_s} A_i B_j \ell(Z_i, f_{S_j}),$$

where $A_i$ is the number of times $Z_i$ appears in the bootstrapped sample $(Z_k^*)_{k=1}^{n_x}$, and $B_j$ is the number of times $S_j$ appears in the bootstrapped sample $(S_k^*)_{k=1}^{n_s}$. With this in mind, we will abuse notation, and also denote $D_n^*$ as the distribution over the $A_i$ and $M_n^*$ as the distribution over the $B_j$. Finally, we will use $\mathbb{E}^*$ and $\text{Var}^*$ to denote the expectation and variance of random variables defined with respect to $D_n^*$ or $M_n^*$, conditional on $D_n$ and $M_n$.

We will use $P$ to denote the distribution $P = D \times M$. Throughout, all assertions made with respect to random variables made without a note about their probability of occurrence hold $P$-almost surely.

*Proof.* The challenge with applying existing theory to our method is that because the performance metric $(\ell(Z_i, f_{S_j})_{i=1}^{n_x}$ over the $n_x$ observations for a given seed $S_j$ all depend on the same $S_j$, they are not independent. Similarly for the performance on a given observation, over seeds. Therefore, we need to handle this non-iid structure in our proof for the multi-bootstrap.

There are conceptually three steps to our proof that allow us to do just that. The first is to show that $\widehat{\theta}$ has an asymptotically linear representation as

$$\sqrt{n}(\widehat{\theta} - \theta) = \sqrt{n}(D_n - D)M\ell + \sqrt{n}(M_n - M)D\ell + o_P(1). \tag{2}$$

The second is to show that conditional on $D_n$ and $M_n$ the multi-bootstrapped statistic $\widehat{\theta}^* \triangleq D_n^* M_n^* \ell$ has an asymptotically linear representation as

$$\sqrt{n}(\widehat{\theta}^* - \widehat{\theta}) = \sqrt{n}(D_n^\circ - D_n)M\ell + \sqrt{n}(M_n^\circ - M_n)D\ell + o_{P^*}(1), \tag{3}$$

where $D_n^\circ$ and $M_n^\circ$ are multiplier bootstrap samples coupled to the bootstrap $D_n^*$ and $M_n^*$ which we define formally in the beginning of Step 2. The third step is to use standard results for the multiplier bootstrap of the mean of iid data to show that the distributions of the above linearized statistics converge to the same limit.

Because we have assumed that $\ell(Z, f_S) < \infty$, $\mathbb{E}[\ell(Z, f_S) \mid S] < \infty$, and $\mathbb{E}[\ell(Z, f_S) \mid Z] < \infty$, Fubini's theorem allows us to switch the order of integration over $Z$ and $S$ as needed.

We will assume that $DM\ell(X, Y, f_S) = 0$. This is without loss of generality, because adding and subtracting $\sqrt{n}DM\ell$ to the bootstrap expression gives

$$\begin{aligned}
\sqrt{n}(\widehat{\theta}^* - \widehat{\theta}) &= \sqrt{n}(D_n^* M_n^* \ell - D_n M_n \ell) \\
&= \sqrt{n}(D_n^* M_n^* \ell - DM\ell + DM\ell - D_n M_n \ell) \\
&= \sqrt{n}(D_n^* M_n^*(\ell - DM\ell) - D_n M_n(\ell - DM\ell)),
\end{aligned}$$

so if we prove that the result holds with the mean zero assumption, it will imply that the result holds for $\ell$ with a nonzero mean.

This theorem guarantees consistency of the Multi-Bootstrap estimates. One question that comes up is whether it is possible to get meaningful / tight rates of convergence for the approximation. Unfortunately, getting $O_P(1/n)$ convergence as found in many bootstrap methods (Van der Vaart, 2000) is difficult without the use of Edgeworth expansions, by which the Multi-Bootstrap is not well-adapted to analysis. That said, many of the remainder terms already have variance of order $O(1/n)$, or could easily be adapted to the same, suggesting an $O_P(1/\sqrt{n})$ convergence. The main difficulty, however, is showing rates of convergence for the strong law on separately exchangeable arrays (see the proof of Lemmas 2, 4-5). Showing a weaker notion of convergence, such as in probability, may perhaps allow one to show that the remainder is $O_P(1/\sqrt{n})$, however the adaptation of the aforementioned Lemmas is nontrivial.

**Step 1**   Recalling that $\widehat{\theta} \triangleq D_n M_n \ell$ and $\theta \triangleq DM\ell$, we can expand $\sqrt{n}(\widehat{\theta} - \theta)$ as follows,

$$
\begin{aligned}
\sqrt{n}(D_n M_n \ell - DM\ell) &= \sqrt{n}(D_n M_n \ell - DM_n \ell + DM_n \ell - DM\ell) \\
&= \sqrt{n}((D_n - D)M_n \ell + D(M_n - M)\ell) \\
&= \sqrt{n}((D_n - D)M_n \ell + (D_n - D)M\ell - (D_n - D)M\ell + D(M_n - M)\ell) \\
&= \sqrt{n}((D_n - D)M\ell + (D_n - D)(M_n - M)\ell + D(M_n - M)\ell)
\end{aligned}
$$

The following lemma shows that $\sqrt{n}(D_n - D)(M_n - M)\ell$ is a lower order term.

**Lemma 1.** *Under the assumptions of Theorem 1,* $\sqrt{n}(D_n - D)(M_n - M)\ell = o_P(1)$.

Therefore,

$$
\sqrt{n}(D_n M_n \ell - DM\ell) = \frac{1}{\sqrt{1 - p_s}}\sqrt{n_x}(D_n - D)M\ell + \frac{1}{\sqrt{p_s}}\sqrt{n_s}(M_n - M)D\ell + o_P(1).
$$

**Step 2**   One of the challenges with working with the bootstrap sample $D_n^*$ and $M_n^*$ is that the induced per-sample weights $\{A_i\}_{i=1}^{n_x}$ and $\{B_j\}_{j=1}^{n_s}$ do not have independent components, because they each follow a multinomial distribution over $n_x$ items and $n_s$ items, respectively. However, they are close enough to independent that we can define a coupled set of random variables $\{A_i^\circ\}_{i=1}^{n_x}$ and $\{B_j^\circ\}_{j=1}^{n_s}$ that do have independent components, but behave similarly enough to $\{A_i\}$ and $\{B_j\}$ that using these weights has a negligible effect on distribution of the bootstrapped estimator, as described concretely below.

First, we discuss the coupled multiplier bootstrap sample $D_n^\circ$ and $M_n^\circ$. The creation of this sequence, called "Poissonization" is a standard technique for proving results about the empirical bootstrap that require independence of the bootstrap weights (van der Vaart et al., 1996). We describe this for $D_n^\circ$ as the idea is identical for $M_n^\circ$. Because our goal is to couple this distribution to $D_n^*$, we define it on the same sample space, and extend the distribution $P^*$, expectation $\mathbb{E}^*$ and variance $\mathrm{Var}^*$ to be over $D_n^\circ$ and $M_n^\circ$, conditionally on $D_n$ and $M_n$, as with $D_n^*$ and $M_n^*$.

To construct the distribution $D_n^\circ$, from the empirical distribution $D_n$ and a bootstrap sample $D_n^*$, start with the distribution $D_n^*$ and modify it as follows: We draw a Poisson random variable $N_{n_x}$ with mean $n_x$. If $N_{n_x} > n_x$, then we sample $N_{n_x} - n_x$ iid observations from $D_n$, with replacement, and add them to the bootstrap sample initialized with $D_n^*$ to produce the distribution $D_n^\circ$. If $N_{n_x} < n_x$, we sample $n_x - N_{n_x}$ observations from $D_n^*$, without replacement, and remove them from the bootstrap sample to produce the distribution $D_n^\circ$. If $N_{n_x} = n_x$, then $D_n^\circ = D_n^*$. Recalling that $A_i$ is the number of times the $i$-th sample is included in $D_n^*$, similarly define $A_i^\circ$ as the number of times the $i$-th sample is included in $D_n^\circ$. Note that by the properties of the Poisson distribution, $A_i^\circ \sim \mathrm{Poisson}(1)$, and $\{A_i^\circ\}_{i=1}^{n_x}$ are independent. Note that the natural normalization for $D_n^\circ$ would be $N_{n_x}$. However, it will be useful to maintain the normalization by $n_x$, so abusing notation, for a function $f(z)$, we will say that $D_n^\circ f = \frac{1}{n_x}\sum_{i=1}^{n_x} A_i^\circ f(Z_i)$.

Define $\widehat{\theta}^\circ$ as the following empirical estimator of $\theta$ under the distribution $D_n^\circ \times M_n^\circ$,

$$\widehat{\theta}^\circ = D_n^\circ M_n^\circ \ell = \frac{1}{n_x} \sum_{i=1}^{n_x} \frac{1}{n_s} \sum_{j=1}^{n_s} A_i^\circ B_j^\circ \ell(Z_i, f_{S_j}).$$

Lemma 2 shows that $\sqrt{n}(\widehat{\theta}^* - \widehat{\theta}^\circ) = o_{P^*}(1)$, and so $\sqrt{n}(\widehat{\theta}^* - \theta) = \sqrt{n}(\widehat{\theta}^\circ - \theta) + o_{P^*}(1)$.

**Lemma 2.** *Under the assumptions of Theorem 1, and that $DM\ell = 0$, $\sqrt{n}(\widehat{\theta}^* - \widehat{\theta}^\circ) = o_{P^*}(1)$.*

With this, the expansion of $\sqrt{n}(\widehat{\theta}^\circ - \widehat{\theta})$ begins *mutatis mutandis* the same as in Step 1, to get that

$$\sqrt{n}(\widehat{\theta}^\circ - \widehat{\theta}) = \frac{1}{\sqrt{1-p_s}} \sqrt{n_x}(D_n^\circ - D_n)M_n\ell + \sqrt{n}(D_n^\circ - D_n)(M_n^\circ - M_n)\ell$$

$$+ \frac{1}{\sqrt{p_s}} \sqrt{n_s}(M_n^\circ - M_n)D_n\ell.$$

As with Step 1, we provide Lemma 3 showing that the remainder term $\sqrt{n}(D_n^\circ - D_n)(M_n^\circ - M_n)\ell$ will be lower order.

**Lemma 3.** *Under the assumptions of Theorem 1, $\sqrt{n}(D_n^\circ - D_n)(M_n^\circ - M_n)\ell = o_{P^*}(1)$.*

Therefore,

$$\sqrt{n}(D_n^\circ M_n^\circ \ell - D_n M_n \ell) = \frac{1}{\sqrt{1-p_s}} \sqrt{n_x}(D_n^\circ - D_n)M_n\ell + \frac{1}{\sqrt{p_s}} \sqrt{n_s}(M_n^\circ - M_n)D_n\ell + o_{P^*}(1).$$

Then, to write $\sqrt{n}(\widehat{\theta}^* - \widehat{\theta})$ in terms of $\sqrt{n_s}(M_n^\circ - M_n)D\ell$ as wanted in Eq. (3), instead of $\sqrt{n_s}(M_n^\circ - M_n)D_n\ell$, we must additionally show that the functional has enough continuity that the error term $\sqrt{n_s}(M_n^\circ - M_n)(D_n - D)\ell$ is lower order. The following lemma shows exactly this.

**Lemma 4.** *Under the assumptions of Theorem 1, conditionally on the sequences $Z_1, Z_2, \ldots$ and $S_1, S_2, \ldots,$*

*(a) $\sqrt{n}(D_n^\circ - D_n)(M_n - M)\ell = o_{P^*}(1)$, and*

*(b) $\sqrt{n}(D_n - D)(M_n^\circ - M_n)\ell = o_{P^*}(1)$.*

Altogether, these imply that

$$\sqrt{n}(D_n^* M_n^* \ell - D_n M_n \ell) = \frac{1}{\sqrt{1-p_s}} \sqrt{n_x}(D_n^\circ - D_n)M\ell + \frac{1}{\sqrt{p_s}} \sqrt{n_s}(M_n^\circ - M_n)D\ell + o_{P^*}(1).$$

$\square$

**Step 3** Noting that $M\ell(\cdot, f_S) = \mathbb{E}_{D \times M}[\ell(\cdot, f_S) \mid Z = \cdot]$ is a real-valued random variable with finite variance (similarly for $D\ell(Z, \cdot)$), and recalling that the $n_x$ samples used for $D_n$ and $n_s$ samples for $M_n$ satisfy $n = n_x/(1-p_s)$ and $n = n_s/p_s$, for $0 < p_s < 1$, the conventional central limit theorem shows that for some positive semi-definite matrix $\Sigma \in \mathbb{R}^{2 \times 2}$, and $G \sim \mathcal{N}(0, \Sigma)$,

$$\sqrt{n} \begin{pmatrix} (D_n - D)M\ell \\ (M_n - M)D\ell \end{pmatrix} = \begin{pmatrix} \frac{1}{1-p_s}\sqrt{n_x}(D_n - D)M\ell \\ \frac{1}{p_s}\sqrt{n_s}(M_n - M)D\ell \end{pmatrix} \xrightarrow{d} G.$$

Note that $D_n$ and $M_n$ are independent, so $G$ is, in fact, a diagonal matrix.

Additionally, the conditional multiplier CLT (van der Vaart et al., 1996, Lemma 2.9.5, pg. 181) implies that conditionally on $Z_1, Z_2, \ldots$ and $S_1, S_2, \ldots,$

$$\sqrt{n} \begin{pmatrix} (D_n^* - D_n)M\ell \\ (M_n^* - M_n)D\ell \end{pmatrix} \xrightarrow{d} G.$$

Finally, applying the delta method (see Theorem 23.5 from Van der Vaart (2000)) along with the results from Steps 1 and 2 shows that the distributions of $\sqrt{n}(\widehat{\theta} - \theta)$ and $\sqrt{n}(\widehat{\theta}^* - \widehat{\theta})$ converge to $\mathcal{N}(0, \sigma^2)$, where $\sigma^2 = \Sigma_{11}/(1-p_s) + \Sigma_{22}/p_s$.

## B.1 PROOF OF LEMMA 1

Fix $\epsilon > 0$. Note that $\mathbb{E}[(D_n - D)(M_n - M)\ell] = 0$, so by Chebyshev's inequality,

$$P\left(|\sqrt{n}(D_n - D)(M_n - M)\ell| > \epsilon\right) \leq \frac{\mathrm{Var}(\sqrt{n}(D_n - D)(M_n - M)\ell)}{\epsilon^2}.$$

Therefore, it suffices to show that $\lim_{n \to \infty} \mathrm{Var}(\sqrt{n}(D_n - D)(M_n - M)\ell) = 0$. To do so, we apply the law of total variance, conditioning on $D_n$, and bound the resulting expression by $C/n$.

$$\mathrm{Var}(\sqrt{n}(D_n - D)(M_n - M)\ell) = n\mathbb{E}[\mathrm{Var}((D_n - D)(M_n - M)\ell \mid D_n)]$$
$$+ n\mathrm{Var}(\mathbb{E}[(D_n - D)(M_n - M)\ell \mid D_n])$$
$$= n\mathbb{E}[\mathrm{Var}((D_n - D)(M_n - M)\ell \mid D_n)]$$
$$= n\mathbb{E}[\mathrm{Var}((M_n - M)(D_n - D)\ell \mid D_n)]$$

$$= \mathbb{E}\left[\frac{n}{n_s^2} \sum_{j=1}^{n_s} \mathrm{Var}((D_n - D)\ell(\cdot, f_{S_j}) \mid D_n)\right]$$

$$= \mathbb{E}\left[\frac{n}{n_s} \mathrm{Var}((D_n - D)\ell(\cdot, f_{S_1}) \mid D_n)\right]$$

$$= \mathbb{E}\left[\frac{1}{p_s} \mathbb{E}\left[\left(\frac{1}{n_x} \sum_{i=1}^{n_x} \ell(Z_i, f_{S_1}) - \mathbb{E}[\ell(Z_i, f_{S_1}) \mid S_1]\right)^2 \mid \{Z_i\}_{i=1}^{n_x}\right]\right]$$

$$= \mathbb{E}\left[\frac{1}{p_s} \left(\frac{1}{n_x} \sum_{i=1}^{n_x} \ell(Z_i, f_{S_1}) - \mathbb{E}[\ell(Z_i, f_{S_1}) \mid S_1]\right)^2\right]$$

$$= \mathbb{E}\left[\frac{1}{p_s n_x^2} \sum_{i=1}^{n_x} \sum_{k=1}^{n_x} (\ell(Z_i, f_{S_1}) - \mathbb{E}[\ell(Z_i, f_{S_1}) \mid S_1])(\ell(Z_k, f_{S_1}) - \mathbb{E}[\ell(Z_k, f_{S_1}) \mid S_1])\right]$$

$$= \mathbb{E}\left[\frac{1}{p_s n_x^2} \sum_{i=1}^{n_x} (\ell(Z_i, f_{S_1}) - \mathbb{E}[\ell(Z_i, f_{S_1}) \mid S_1])^2\right]$$

$$= \frac{1}{p_s(1 - p_s)n} \mathbb{E}\left[(\ell(Z_1, f_{S_1}) - \mathbb{E}[\ell(Z_1, f_{S_1}) \mid S_1])^2\right] \leq \frac{C}{n} \to 0.$$

## B.2 PROOF OF LEMMA 2

First, note the following representation for $\widehat{\theta}^* - \widehat{\theta}^\circ$:

$$\widehat{\theta}^* - \widehat{\theta}^\circ = \frac{1}{n_x} \sum_{i=1}^{n_x} \frac{1}{n_s} \sum_{j=1}^{n_s} A_i B_j \ell(Z_i, f_{S_j}) - \frac{1}{n_x} \sum_{i=1}^{n_x} \frac{1}{n_s} \sum_{j=1}^{n_s} A_i^\circ B_j^\circ \ell(Z_i, f_{S_j})$$

$$= \underbrace{\frac{1}{n_s} \sum_{j=1}^{n_s} \frac{(B_j - B_j^\circ)}{n_x} \sum_{i=1}^{n_x} A_i \ell(Z_i, f_{S_j})}_{\triangleq I_1} + \underbrace{\frac{1}{n_x} \sum_{i=1}^{n_x} \frac{(A_i - A_i^\circ)}{n_s} \sum_{j=1}^{n_s} B_j^\circ \ell(Z_i, f_{S_j})}_{\triangleq I_2}.$$

Let $\epsilon > 0$. Noting that $\mathbb{E}^*[I_1] = \mathbb{E}^*[I_2] = 0$, applying Chebyshev's inequality gives

$$P^*\left(\sqrt{n}|\widehat{\theta}^* - \widehat{\theta}^\circ| > \epsilon\right) \leq n\frac{\mathrm{Var}^*(\widehat{\theta}^* - \widehat{\theta}^\circ)}{\epsilon^2} \leq 2n\frac{\mathrm{Var}^*(I_1) + \mathrm{Var}^*(I_2)}{\epsilon^2}$$

It suffices to show that $n\mathrm{Var}^*(I_1) \to 0$ and $n\mathrm{Var}^*(I_2) \to 0$. The arguments for each term are *mutatis mutandis* the same, and so we proceed by showing the proof for $I_2$.

By the law of total variance,

$$\mathrm{Var}^*(I_2) = \mathrm{Var}^*(\mathbb{E}^*[I_2 \mid \{B_j\}_{j=1}^{n_s}]) + \mathbb{E}^*[\mathrm{Var}^*(I_2 \mid \{B_j\}_{j=1}^{n_s})].$$

Because $\mathbb{E}^*[A_i] = \mathbb{E}^*[A_i^\circ]$ and $\{B_j\}_{j=1}^{n_s} \perp\!\!\!\perp A_i, A_i^\circ$, it follows that $\mathbb{E}^*[I_2 \mid \{B_j\}_{j=1}^{n_s}] = 0$. Taking the remaining term and re-organizing the sums in $I_2$,

$$\text{Var}^*(I_2) = \mathbb{E}^* \left[ \text{Var}^* \left( \frac{1}{n_x} \sum_{i=1}^{n_x} (A_i - A_i^\circ) \left[ \frac{1}{n_s} \sum_{j=1}^{n_s} B_j \ell(Z_i, f_{S_j}) \right] \mid \{B_j\}_{j=1}^{n_s} \right) \right]. \quad (4)$$

Next, we apply the law of total variance again, conditioning on $N_{n_x} = \sum_i A_i^\circ$. First,

$$\mathbb{E}^*[I_2 \mid N_{n_x}, \{B_j\}_{j=1}^{n_s}] = \frac{N_{n_x} - n_x}{n_x} \frac{1}{n_x} \sum_{i=1}^{n_x} \frac{1}{n_s} \sum_{j=1}^{n_s} B_j \ell(Z_i, f_{S_j}),$$

and so

$$\text{Var}^* \left( \mathbb{E}^*[I_2 \mid N_{n_x}, \{B_j\}_{j=1}^{n_s}] \mid \{B_j\}_{j=1}^{n_s} \right) = \frac{1}{n_x} \left( \frac{1}{n_x} \sum_{i=1}^{n_x} \frac{1}{n_s} \sum_{j=1}^{n_s} B_j \ell(Z_i, f_{S_j}) \right)^2$$

Then, conditionally on $N_{n_x}$ (and $\{B_j\}$), $I_2$ is the (centered) empirical average of $|N_n - n|$ samples from a finite population of size $n$, rescaled by $|N_n - n|/n$. Therefore, applying Theorem 2.2 of Cochran (2007) gives the conditional variance as

$$\frac{|N_{n_x} - n_x|}{n_x^2} \underbrace{\left( \frac{1}{n_x - 1} \sum_{i=1}^{n_x} \left[ \frac{1}{n_s} \sum_{j=1}^{n_s} B_j \ell(Z_i, f_{S_j}) \right]^2 - \frac{n_x}{n_x - 1} \left[ \frac{1}{n_x} \sum_{i=1}^{n_x} \frac{1}{n_s} \sum_{j=1}^{n_s} B_j \ell(Z_i, f_{S_j}) \right]^2 \right)}_{\triangleq V^2}.$$

To take the expectation over $N_{n_x}$, notice that because $\mathbb{E}^*[N_{n_x}] = n_x$, this is the mean absolute deviation (MAD) of $N_{n_x}$. Using the expression for the MAD of a Poisson variable from Ramasubban (1958) gives

$$\mathbb{E}^*|N_{n_x} - n_x| = 2n_x \frac{n_x^{n_x} \exp(-n_x)}{n_x!},$$

and using Stirling's approximation, this is bounded by $C\sqrt{n_x}$, for some $0 < C < \infty$.

Combining this with the above term for the variance of the conditional expectation, we have

$$\text{Var}^* \left( \frac{1}{n_x} \sum_{i=1}^{n_x} (A_i - A_i^\circ) \left[ \frac{1}{n_s} \sum_{j=1}^{n_s} B_j \ell(Z_i, f_{S_j}) \right] \mid \{B_j\}_{j=1}^{n_s} \right) \leq \frac{1}{n_x} \left( \frac{1}{n_x} \sum_{i=1}^{n_x} \frac{1}{n_s} \sum_{j=1}^{n_s} B_j \ell(Z_i, f_{S_j}) \right)^2$$

$$+ \frac{1}{n_x^{1.5}} V^2. \quad (5)$$

Noting that $\mathbb{E}^*[B_j^2] = \mathbb{E}^*[B_j B_k] = 1$, we get the following bound:

$$\text{Var}^*(I_2) \leq \frac{1}{n_x} \left( \frac{1}{n_x} \sum_{i=1}^{n_x} \frac{1}{n_s} \sum_{j=1}^{n_s} \ell(Z_i, f_{S_j}) \right)^2 + \frac{1}{n_x^{1.5}} \bar{V}^2,$$

where

$$\bar{V}^2 = \frac{1}{n_x - 1} \sum_{i=1}^{n_x} \left[ \frac{1}{n_s} \sum_{j=1}^{n_s} \ell(Z_i, f_{S_j}) \right]^2 - \frac{n_x}{n_x - 1} \left[ \frac{1}{n_x} \sum_{i=1}^{n_x} \frac{1}{n_s} \sum_{j=1}^{n_s} \ell(Z_i, f_{S_j}) \right]^2.$$

Because of the assumption that $DM\ell = 0$, the SLLN adapted to separately exchangeable arrays (Rieders, 1991, Theorem 1.4) implies that

$$\lim_{n \to \infty} \frac{1}{n_x} \sum_{i=1}^{n_x} \frac{1}{n_s} \sum_{j=1}^{n_s} \ell(Z_i, f_{S_j}) = 0,$$

almost surely. Therefore, the first term of (5) is $o(1/n)$.

Note that $\bar{V}^2$ is the empirical variance of the conditional expectation of $\ell(Z_i, f_{S_j})$ given $\{Z_i\}_{i=1}^n$. Therefore, the law of total variance shows that

$$\bar{V}^2 \leq \frac{1}{n_x}\frac{1}{n_s}\sum_{i=1}^{n_x}\sum_{j=1}^{n_s}\ell^2(Z_i, f_{S_j}) - \left(\frac{1}{n_x}\frac{1}{n_s}\sum_{i=1}^{n_x}\sum_{j=1}^{n_s}\ell(Z_i, f_{S_j})\right)^2.$$

By the SLLN adapted to separately exchangeable arrays (Rieders, 1991, Theorem 1.4), both of the terms converge almost surely to $DM\ell^2 < \infty$ and $(DM\ell)^2$, respectively. and therefore,

$$\lim_{n\to\infty} n\mathrm{Var}^*(I_s) \leq \lim_{n\to\infty}\frac{n}{n_x}\left(\frac{1}{n_x}\sum_{i=1}^{n_x}\frac{1}{n_s}\sum_{j=1}^{n_s}\ell(Z_i, f_{S_j})\right)^2 + \frac{n}{n_x^{1.5}}\bar{V}^2 = 0.$$

### B.3 PROOF OF LEMMA 3

As with Lemma 1, the main idea of the proof is to apply Chebyshev's inequality, and show that the variance tends to zero. Indeed, choosing an arbitrary $\epsilon > 0$,

$$P^*\left(|\sqrt{n}(D_n^\circ - D_n)(M_n^\circ - M_n)\ell| \geq \epsilon\right) \leq \frac{\mathrm{Var}^*\left(\sqrt{n}(D_n^\circ - D_n)(M_n^\circ - M_n)\ell\right)}{\epsilon^2}.$$

Therefore, it suffices to show that the variance in the above display goes to zero. To do this, we start by re-writing the expression in terms of $A_i^\circ$ and $B_j^\circ$, and then apply the law of total variance.

$$\mathrm{Var}^*\left(\sqrt{n}(D_n^\circ - D_n)(M_n^\circ - M_n)\ell\right) = n\mathrm{Var}^*\left(\frac{1}{n_x n_s}\sum_{i=1}^{n_x}\sum_{j=1}^{n_s}(A_i^\circ - 1)(B_j^\circ - 1)\ell(Z_i, f_{S_j})\right)$$

$$= n\mathrm{Var}^*\left(\mathbb{E}^*\left[\frac{1}{n_x n_s}\sum_{i=1}^{n_x}\sum_{j=1}^{n_s}(A_i^\circ - 1)(B_j^\circ - 1)\ell(Z_i, f_{S_j}) \mid \{A_i^\circ\}_{i=1}^{n_x}\right]\right)$$

$$+ n\mathbb{E}^*\left[\mathrm{Var}^*\left(\frac{1}{n_x n_s}\sum_{i=1}^{n_x}\sum_{j=1}^{n_s}(A_i^\circ - 1)(B_j^\circ - 1)\ell(Z_i, f_{S_j}) \mid \{A_i^\circ\}_{i=1}^{n_x}\right)\right].$$

Because $\{B_j^\circ\}_{j=1}^{n_s}$ are independent of $\{A_i^\circ\}_{i=1}^{n_x}$, and have mean 1, the conditional expectation in the first term is 0 almost surely. Expanding out the second term, using that $\mathrm{Var}^*(B_j^\circ) = 1$, and that the $\{B_j^\circ\}_{j=1}^{n_s}$ are uncorrelated,

$$n\mathbb{E}^*\left[\mathrm{Var}^*\left(\frac{1}{n_x n_s}\sum_{i=1}^{n_x}\sum_{j=1}^{n_s}(A_i^\circ - 1)(B_j^\circ - 1)\ell(Z_i, f_{S_j}) \mid \{A_i\}_{i=1}^{n_x}\right)\right]$$

$$= n\mathbb{E}^*\left[\frac{1}{n_s^2}\sum_{j=1}^{n_s}\mathrm{Var}^*\left((B_j^\circ - 1)\frac{1}{n_x}\sum_{i=1}^{n_x}(A_i^\circ - 1)\ell(Z_i, f_{S_j}) \mid \{A_i^\circ\}_{i=1}^{n_x}\right)\right]$$

$$= n\mathbb{E}^*\left[\frac{1}{n_s^2}\sum_{j=1}^{n_s}\left(\frac{1}{n_x}\sum_{i=1}^{n_x}(A_i^\circ - 1)\ell(Z_i, f_{S_j})\right)^2\right]$$

$$= n\mathbb{E}^*\left[\frac{1}{n_s^2}\sum_{j=1}^{n_s}\frac{1}{n_x^2}\sum_{i=1}^{n_x}\sum_{k=1}^{n_x}(A_i^\circ - 1)(A_k^\circ - 1)\ell(Z_i, f_{S_j})\ell(Z_k, f_{S_j})\right].$$

Now, noting that $\text{Var}^*(A_i^\circ) = 1$, and that the $\{A_i^\circ\}_{i=1}^{n_x}$ are uncorrelated, this simplifies to

$$n\mathbb{E}^* \left[ \frac{1}{n_s^2} \sum_{j=1}^{n_s} \frac{1}{n_x^2} \sum_{i=1}^{n_x} (A_i^\circ - 1)^2 \ell^2(Z_i, f_{S_j}) \right] = \frac{n}{n_s n_x} \frac{1}{n_s} \sum_{j=1}^{n_s} \frac{1}{n_x} \sum_{i=1}^{n_x} \ell^2(Z_i, f_{S_j}).$$

Because $\mathbb{E}_{D \times M}[\ell^2(Z, f_S)] < \infty$, the SLLN adapted to separately exchangeable arrays (Rieders, 1991, Theorem 1.4) implies that this converges almost surely to 0.

## B.4 PROOF OF LEMMA 4

We prove (a) of the Lemma, as (b) follows from applying Fubini's theorem and following *mutatis mutandis* the same argument. Without loss of generality, we will assume that $\ell(Z_i, f_{S_j}) \geq 0$. Because $\text{Var}(\ell(Z_i, f_{S_j})) < \infty$, we can always decompose $\ell(\cdot, \cdot)$ into a positive and negative part, and show that the result holds for each individually.

Once again, we prove (a) by turning to Chebyshev's inequality. Fix $\epsilon > 0$, and observe that

$$P^* \left( |\sqrt{n}(D_n^\circ - D_n)(M_n - M)\ell| > \epsilon \right) \leq \frac{\text{Var}^* \left( \sqrt{n}(D_n^\circ - D_n)(M_n - M) \right)}{\epsilon^2},$$

so it is sufficient to show that $\text{Var}^* \left( \sqrt{n}(D_n^\circ - D_n)(M_n - M) \right) \to 0$.

Writing the above in terms of $A_i^\circ$, we have

$$\text{Var}^* \left( \sqrt{n}(D_n^\circ - D_n)(M_n - M) \right)$$

$$= \text{Var}^* \left( \frac{\sqrt{n}}{n_x} \sum_{i=1}^{n_x} (A_i^\circ - 1) \left( \frac{1}{n_s} \sum_{j=1}^{n_s} \ell(Z_i, f_{S_j}) - \mathbb{E}[\ell(Z_i, f_{S_j}) \mid Z_i] \right) \right)$$

$$= \frac{n}{n_x^2} \sum_{i=1}^{n_x} \text{Var}^* (A_i^\circ - 1) \left( \frac{1}{n_s} \sum_{j=1}^{n_s} \ell(Z_i, f_{S_j}) - \mathbb{E}[\ell(Z_i, f_{S_j}) \mid Z_i] \right)^2$$

$$= \frac{n}{n_x^2} \sum_{i=1}^{n_x} \left( \frac{1}{n_s} \sum_{j=1}^{n_s} \ell(Z_i, f_{S_j}) - \mathbb{E}[\ell(Z_i, f_{S_j}) \mid Z_i] \right)^2.$$

Now, we want to show that the last display converges almost surely to 0. Notice that each term within the outer sum will obviously converge due to the SLLN. Showing that the outer sum also converges almost surely is technically difficult, but conceptually follows the same argument used to prove the SLLN (specifically, we follow the one done elegantly by Etemadi (1981); Luzia (2018) provides a more detailed account of this proof technique that is helpful for developing a deeper understanding).

We show the following version of almost sure convergence: that for any $\epsilon > 0$,

$$P \left( \frac{n}{n_x^2} \sum_{i=1}^{n_x} \left( \frac{1}{n_s} \sum_{j=1}^{n_s} \ell(Z_i, f_{S_j}) - \mathbb{E}[\ell(Z_i, f_{S_j}) \mid S_j] \right)^2 > \epsilon \text{ i.o.} \right) = 0,$$

where i.o. stands for infinitely often.

Define the shorthand $L_{ij} = \ell(Z_i, f_{S_j})$ and let $\bar{L}_{ij} = L_{ij} 1\{L_{ij} < ij\}$ be a truncated version of $L_{ij}$. The proof of Theorem 2 of Etemadi (1981) implies that $P(\bar{L}_{ij} \neq L_{ij} \text{ i.o.}) = 0$, because the assumption $\text{Var}(L_{ij}) < \infty$ implies the assumption used in Etemadi (1981), and independence of $\{L_{ij}\}_{i,j}$ is not needed for this result. Therefore,

$$\frac{1}{n_x} \sum_{i=1}^{n_x} \left( \frac{1}{n_s} \sum_{j=1}^{n_s} L_{ij} - \bar{L}_{ij} \right)^2 \overset{a.s.}{\to} 0, \text{ and } \frac{1}{n_x} \sum_{i=1}^{n_x} \left( \frac{1}{n_s} \sum_{j=1}^{n_s} \mathbb{E}[L_{ij} \mid Z_i] - \mathbb{E}[\bar{L}_{ij} \mid Z_i] \right)^2 \overset{a.s.}{\to} 0.$$

Together, these imply that if we can prove that the truncated sum converges, ie.,

$$\frac{1}{n_x} \sum_{i=1}^{n} \left( \frac{1}{n_s} \sum_{j=1}^{n_s} \bar{L}_{ij} - \mathbb{E}[\bar{L}_{ij} \mid Z_i] \right)^2 \overset{a.s.}{\to} 0, \tag{6}$$

this is sufficient to show that the un-truncated version converges almost surely.

To prove (6), we show two things: first, that there is a subsequence $k_n$ such that (6) holds when restricted to the subsequence, and then we show that the sequence is a Cauchy sequence, which together imply the result.

Let $\alpha > 1$ and let $k_n = \alpha^n$. For convenience, denote $k_{nx}$ as the number of data samples and $k_{ns}$ as the number of seed samples when $k_{nx} + k_{ns} = k_n$ total samples are drawn. We will ignore integer rounding issues, and assume $k_{nx} = (1 - p_s)\alpha^n$, and $k_{ns} = p_s \alpha^n$.

The following lemma shows that the subsequence defined by $k_n$ converges almost surely.

**Lemma 5.** *Let $\alpha > 1$, and $k_n = \alpha^n$. Under the assumptions of Theorem 1 and that $L_{ij} \geq 0$*

$$P \left( \frac{1}{k_{nx} k_{ns}^2} \sum_{i=1}^{k_{nx}} \left( \sum_{j=1}^{k_{ns}} \bar{L}_{ij} - \mathbb{E}[\bar{L}_{ij} \mid Z_i] \right)^2 > \epsilon \; i.o. \right) = 0.$$

We now must show that the sequence in (6) is a Cauchy sequence. Note that the SLLN implies that

$$\frac{1}{n_x} \sum_{i=1}^{n_x} \mathbb{E}[\bar{L}_{ij} \mid Z_i]^2 \overset{a.s.}{\to} \mathbb{E}[\mathbb{E}[\bar{L}_{ij} \mid Z_i]^2],$$

and the LLN for exchangeable arrays (Rieders, 1991, Theorem 1.4) implies that

$$\frac{1}{n_x} \sum_{i=1}^{n_x} \frac{1}{n_s} \sum_{j=1}^{n_s} \bar{L}_{ij} \mathbb{E}[\bar{L}_{ij} \mid Z_i] \overset{a.s.}{\to} \mathbb{E}[\mathbb{E}[\bar{L}_{ij} \mid Z_i]^2].$$

Therefore,

$$\frac{1}{k_{nx} k_{ns}^2} \sum_{i=1}^{k_{nx}} \left( \sum_{j=1}^{k_{ns}} \bar{L}_{ij} \right)^2 \overset{a.s.}{\to} \mathbb{E}[\mathbb{E}[\bar{L}_{ij} \mid Z_i]^2]. \tag{7}$$

Notice that because $\bar{L}_{ij} \geq 0$, the sum $\sum_{i=1}^{n_x} \left( \sum_{j=1}^{n_s} \bar{L}_{ij} \right)^2$ is monotone increasing in $n_s$ and $n_x$. With this in mind, for any $m > 0$, let $n$ be such that $k_n \leq m < k_{n+1}$. Then, by the montonicity,

$$\left( \frac{k_n}{k_{n+1}} \frac{1}{k_n} \right)^3 \sum_{i=1}^{k_{nx}} \left( \sum_{j=1}^{k_{ns}} \bar{L}_{ij} \right)^2 \leq \frac{\sum_{i=1}^{(1-p_s)m} \left( \sum_{j=1}^{p_s m} \bar{L}_{ij} \right)^2}{p_s^2 (1 - p_s) m^3} \leq \left( \frac{k_{n+1}}{k_n} \frac{1}{k_{n+1}} \right)^3 \sum_{i=1}^{k_{(n+1)x}} \left( \sum_{j=1}^{k_{(n+1)s}} \bar{L}_{ij} \right)^2.$$

From (7), the left hand side converges to $\frac{1}{\alpha^3} \mathbb{E}[\mathbb{E}[\bar{L}_{ij} \mid Z_i]^2]$, and the right hand side converges to $\alpha^3 \mathbb{E}[\mathbb{E}[\bar{L}_{ij} \mid Z_i]^2]$. Because $\alpha$ is arbitrary, this proves that the sequence

$$\left( \frac{\sum_{i=1}^{(1-p_s)m} \left( \sum_{j=1}^{p_s m} \bar{L}_{ij} \right)^2}{p_s^2 (1 - p_s) m^3} \right)_{m=1,\dots}$$

is almost surely Cauchy. Together with Lemma 5, this implies (6).

### B.5 PROOF OF LEMMA 5

We will show that

$$\sum_{n=1}^{\infty} P \left( \frac{1}{k_{nx} k_{ns}^2} \sum_{i=1}^{k_{nx}} \left( \sum_{j=1}^{k_{ns}} \bar{L}_{ij} - \mathbb{E}[\bar{L}_{ij} \mid Z_i] \right)^2 > \epsilon \right) < \infty.$$

This, along with the first Borel-Cantelli lemma (Émile Borel, 1909; Cantelli, 1917) implies the result.

Applying Markov's inequality and using the fact that $\bar{L}_{ij}$ and $\bar{L}_{ih}$ are independent conditional on $Z_i$ gives

$$\sum_{n=1}^{\infty} P\left(\frac{1}{k_{nx}k_{ns}^2}\sum_{i=1}^{k_{nx}}\left(\sum_{j=1}^{k_{ns}}\bar{L}_{ij} - \mathbb{E}[\bar{L}_{ij} \mid Z_i]\right)^2 > \epsilon\right)$$

$$\leq \frac{1}{\epsilon}\sum_{n=1}^{\infty}\mathbb{E}\left[\frac{1}{k_{nx}k_{ns}^2}\sum_{i=1}^{k_{nx}}\left(\sum_{j=1}^{k_{ns}}\bar{L}_{ij} - \mathbb{E}[\bar{L}_{ij} \mid Z_i]\right)^2\right]$$

$$= \frac{1}{\epsilon}\sum_{n=1}^{\infty}\frac{1}{k_{nx}k_{ns}^2}\sum_{i=1}^{k_{nx}}\sum_{j=1}^{k_{ns}}\mathbb{E}\left[\left(\bar{L}_{ij} - \mathbb{E}[\bar{L}_{ij} \mid Z_i]\right)^2\right]$$

$$\leq \frac{1}{\epsilon}\sum_{n=1}^{\infty}\frac{1}{k_{nx}k_{ns}^2}\sum_{i=1}^{k_{nx}}\sum_{j=1}^{k_{ns}}\mathbb{E}[\bar{L}_{ij}^2],$$

where the last line follows from the law of total variance. To simplify the remaining algebra, we will use $a \lesssim b$ to denote that there is some constant $0 < c < \infty$ such that $a < cb$. Continuing, we have

$$\frac{1}{\epsilon}\sum_{n=1}^{\infty}\frac{1}{k_{nx}k_{ns}^2}\sum_{i=1}^{k_{nx}}\sum_{j=1}^{k_{ns}}\mathbb{E}[\bar{L}_{ij}^2] \lesssim \frac{1}{\epsilon}\sum_{n=1}^{\infty}\sum_{i=1}^{k_{nx}}\sum_{j=1}^{k_{ns}}\frac{1}{k_n^3}\mathbb{E}[\bar{L}_{ij}^2]$$

$$= \frac{1}{\epsilon}\sum_{i=1}^{\infty}\sum_{j=1}^{\infty}\mathbb{E}[\bar{L}_{ij}^2]\sum_{n=n(i,j)}^{\infty}\frac{1}{\alpha^{3n}}$$

$$\lesssim \frac{1}{\epsilon}\sum_{i=1}^{\infty}\sum_{j=1}^{\infty}\frac{1}{\max\{i/(1-p_s), j/p_s\}^3}\mathbb{E}[\bar{L}_{ij}^2]$$

$$\lesssim \frac{1}{\epsilon}\sum_{i=1}^{\infty}\sum_{j=1}^{\infty}\frac{1}{\max\{i, j\}^3}\mathbb{E}[\bar{L}_{ij}^2]$$

$$= \frac{1}{\epsilon}\sum_{i=1}^{\infty}\sum_{j=1}^{\infty}\frac{1}{\max\{i, j\}^3}\mathbb{E}[\bar{L}_{ij}^2]$$

where $n(i, j)$ is shorthand for $n(i, j) = \log_\alpha \max\{i/(1-p_s), j/p_s\}$ is the first $n$ such that $k_{nx} \geq i$ and $k_{ns} \geq j$.

Now, define $Q$ as the distribution of $L_{11}$ induced by $Z_1$ and $S_1$. Additionally, split the inner sum into two pieces, one for when $j < i$ and so $\max\{i, j\} = i$ and one for when $j \geq i$ and so $\max\{i, j\} = j$.

$$\frac{1}{\epsilon}\sum_{i=1}^{\infty}\sum_{j=1}^{\infty}\frac{1}{\max\{i, j\}^3}\mathbb{E}[\bar{L}_{ij}^2] = \frac{1}{\epsilon}\sum_{i=1}^{\infty}\left(\sum_{j=1}^{i}\frac{1}{i^3}\int_0^{ij}x^2\,\mathrm{d}Q(x) + \sum_{j=i}^{\infty}\int_0^{ij}x^2\,\mathrm{d}Q(x)\right)$$

$$= \frac{1}{\epsilon}\sum_{i=1}^{\infty}\left(\sum_{j=1}^{i-1}\frac{1}{i^3}\sum_{k=1}^{ij}\int_{k-1}^{k}x^2\,\mathrm{d}Q(x) + \sum_{j=i}^{\infty}\sum_{k=1}^{ij}\int_{k-1}^{k}x^2\,\mathrm{d}Q(x)\right)$$

switching the order of the indices over $j$ and $k$, using that $1 \leq k \leq ij$ and the constraints on $j$ relative to $i$,

$$\frac{1}{\epsilon} \sum_{i=1}^{\infty} \left( \sum_{j=1}^{i-1} \frac{1}{i^3} \sum_{k=1}^{ij} \int_{k-1}^{k} x^2 \, \mathrm{d}Q(x) + \sum_{j=i}^{\infty} \sum_{k=1}^{ij} \int_{k-1}^{k} x^2 \, \mathrm{d}Q(x) \right)$$

$$\lesssim \frac{1}{\epsilon} \sum_{i=1}^{\infty} \left( \sum_{k=1}^{i^2-1} \frac{(i-k/i)}{i^3} \int_{k-1}^{k} x^2 \, \mathrm{d}Q(x) + \sum_{k=1}^{\infty} \sum_{j=\max\{i,k/i\}}^{\infty} \frac{1}{j^3} \int_{k-1}^{k} x^2 \, \mathrm{d}Q(x) \right)$$

$$\lesssim \frac{1}{\epsilon} \sum_{i=1}^{\infty} \left( \sum_{k=1}^{i^2-1} \frac{(i-k/i)}{i^3} \int_{k-1}^{k} x^2 \, \mathrm{d}Q(x) + \sum_{k=1}^{\infty} \frac{1}{\max\{i,k/i\}^2} \int_{k-1}^{k} x^2 \, \mathrm{d}Q(x) \right).$$

Switching the order of summation over $i$ and $k$, and separating out the terms where $k/i < i$ and $k/i \geq i$,

$$\frac{1}{\epsilon} \sum_{i=1}^{\infty} \left( \sum_{k=1}^{i^2-1} \frac{(i-k/i)}{i^3} \int_{k-1}^{k} x^2 \, \mathrm{d}Q(x) + \sum_{k=1}^{\infty} \frac{1}{\max\{i,k/i\}^2} \int_{k-1}^{k} x^2 \, \mathrm{d}Q(x) \right)$$

$$= \frac{1}{\epsilon} \sum_{k=1}^{\infty} \left( \int_{k-1}^{k} x^2 \, \mathrm{d}Q(x) \right) \left( \sum_{i=1}^{\sqrt{k}+1} \frac{(i-k/i)}{i^3} + \sum_{i=\sqrt{k}}^{\infty} \frac{1}{i^2} + \sum_{i=1}^{\sqrt{k}} \frac{i^2}{k^2} \right)$$

$$\lesssim \frac{1}{\epsilon} \sum_{k=1}^{\infty} \frac{1}{\sqrt{k}} \left( \int_{k-1}^{k} x^2 \, \mathrm{d}Q(x) \right)$$

$$\lesssim \frac{1}{\epsilon} \sum_{k=1}^{\infty} \left( \int_{k-1}^{k} \frac{x^2}{\sqrt{x}} \, \mathrm{d}Q(x) \right)$$

$$\lesssim \frac{1}{\epsilon} \int_{0}^{\infty} x^{1.5} \, \mathrm{d}Q(x) < \infty.$$

## C  INSTANCE-LEVEL AGREEMENT OF MULTIBERTS ON GLUE

We present additional performance experiments to complement Section 2.

Table 3 shows per-example agreement rates on GLUE predictions between pairs of models pre-trained with a single seed ("same") and pairs pre-trained with different seeds ("diff"); in all cases, models are fine-tuned with different seeds. With the exception of RTE, we see high agreement (over 90%) on test examples drawn from the same distribution as the training data, and note that agreement is 1–2% lower on average for the predictions of models pre-trained on different seeds compared to models pre-trained on the same seed. However, this discrepancy becomes significantly more pronounced if we look at out-of-domain "challenge sets" which feature a different data distribution from the training set. For example, if we evaluate our MNLI models on the anti-sterotypical examples from HANS (McCoy et al., 2019), we see agreement drop from 88% to 82% when comparing across pre-training seeds. Figure 4 shows how this can affect overall accuracy, which can vary over a range of nearly 20% depending on the pre-training seed. Such results underscore the need to evaluate multiple pre-training runs, especially when evaluating a model's ability to generalize outside of its training distribution.

|  | Same | Diff. | Same - Diff. |
|---|---|---|---|
| CoLA | 91.5% | 89.7% | 1.7% |
| MNLI | 93.6% | 90.1% | 3.5% |
| *HANS (all)* | *92.2%* | *88.1%* | *4.1%* |
| *HANS (neg)* | *88.3%* | *81.9%* | *6.4%* |
| MRPC | 91.7% | 90.4% | 1.3% |
| QNLI | 95.0% | 93.2% | 1.9% |
| QQP | 95.0% | 94.1% | 0.9% |
| RTE | 74.3% | 73.0% | 1.3% |
| SST-2 | 97.1% | 95.6% | 1.4% |
| STS-B | 97.6% | 96.2% | 1.4% |

Table 3: Average per-example agreement between model predictions on each task. This is computed as the average "accuracy" between the predictions of two runs for classification tasks, or Pearson correlation for regression (STS-B). We separate pairs of models that use the same pre-training seed but different fine-tuning seeds (Same) and pairs that differ both in their pre-training and fine-tuning seeds (Diff). HANS (neg) refers to only the anti-stereotypical examples (non-entailment), which exhibit significant variability between models (McCoy et al., 2020).

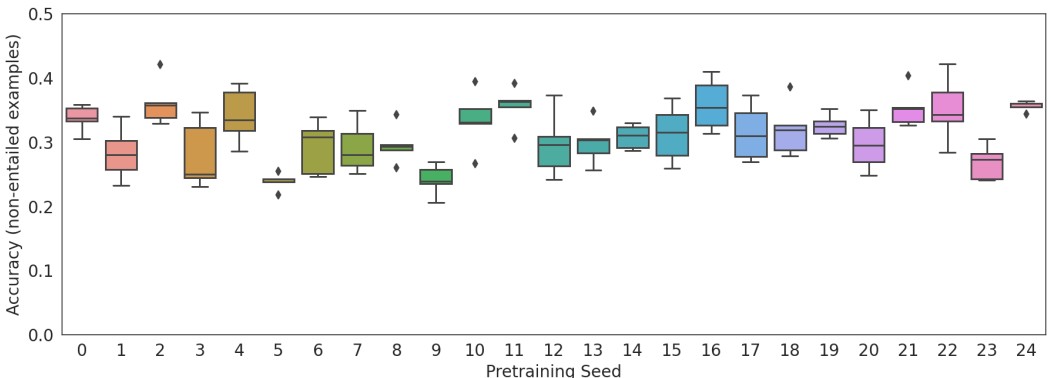

Figure 4: Accuracy of MNLI models on the anti-stereotypical (non-entailment) examples from HANS (McCoy et al., 2020), grouped by pre-training seed. Each column shows the distribution of five fine-tuning runs based on the same initial checkpoint.

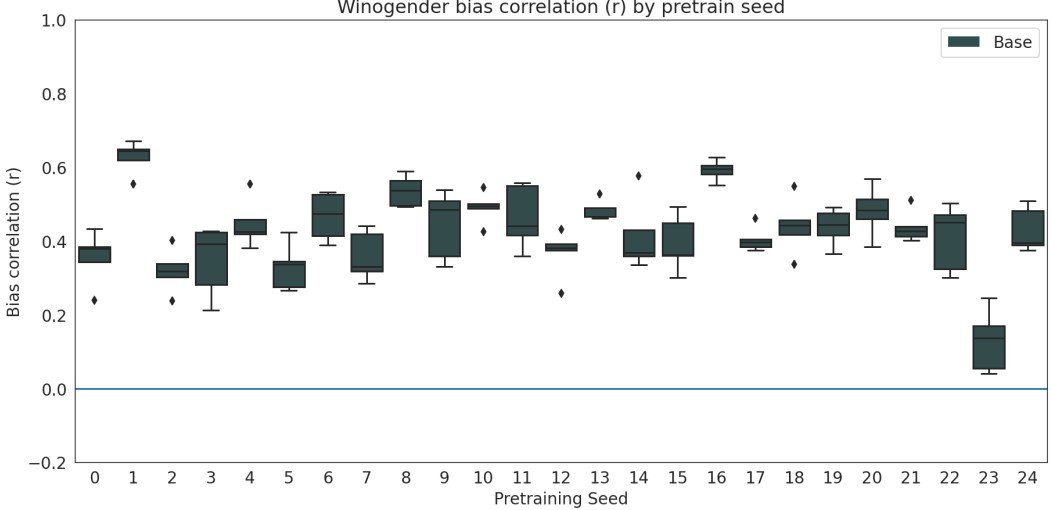

Figure 5: Bias correlation on Winogender for each pre-training seed. Each box represents the distribution of the score over five training runs of the coreference model over each MultiBERTs base checkpoint. This is the same data as Figure 3, but showing only the base checkpoints.

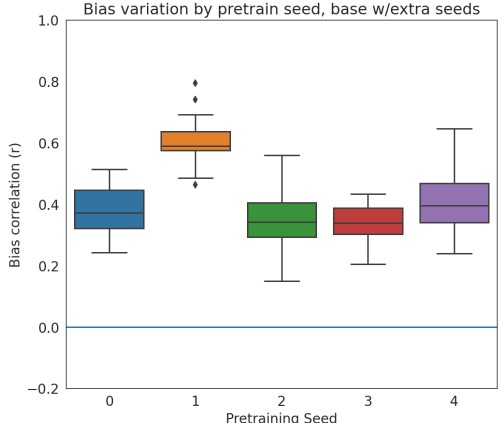

| | | **Bias** ($r$) |
|---|---|---|
| Seed 0 | $\theta_f$ | 0.368 |
| Seed 1 | $\theta_{f'}$ | 0.571 |
| Avg. Diff. | $\delta = \theta_{f'} - \theta_f$ | 0.203 |
| p-value | | 0.009 |

Figure 6: Bias correlation on Winogender for five pretraining seeds, with 25 coreference runs per seed.

Table 4: Unpaired Multi-Bootstrap on Winogender bias correlation, comparing pretraining seed 0 to pretraining seed 1.

## D  CROSS-SEED VARIATION

Figure 5 shows variation in Winogender bias correlation (S4) between each MultiBERTs pretraining seed. Each box shows the distribution over five runs, and some of the variation between *seeds* may simple be due to variation in training the coreference model. If we average the scores for each seed then look at the distribution of this per-seed average score, we get 0.45±0.11. What if pretraining didn't matter? If we ignore the seed and randomly sample sets of five runs from this set with replacement, we get scores of 0.45±0.05 - telling us that most of the variance can only be explained by differences between the pretraining checkpoints.

We can confirm this by taking a subset of our pretraining seeds and training additional 25 randomly-initialized coreference models. Figure 6 shows the result: seeds 0, 2, 3, and 4 appear closer together than in Figure 5, but seed 1 clearly has different properties with respect to our Winogender metric. We can confirm this with an unpaired multibootstrap analysis, taking seed 0 as base and seed 1 as experiment: we observe a significant effect of $\delta = 0.203$ ($p = 0.009$), as shown in Table 4.

|  | **MNLI** | **RTE** | **MRPC** |
|---|---|---|---|
| $\theta_f$ (1M steps) | 0.837 | 0.644 | 0.861 |
| $\theta_{f'}$ (2M steps) | 0.844 | 0.655 | 0.860 |
| $\delta = \theta_{f'} - \theta_f$ | 0.007 | 0.011 | -0.001 |
| $p$-value ($H_0$ that $\delta \leq 0$) | $< 0.001$ | 0.141 | 0.564 |

Table 5: Expected scores (accuracy), effect sizes, and $p$-values from Multi-Bootstrap on selected GLUE tasks. We pre-select the best fine-tuning learning rate by averaging over runs; this is 3e-5 for checkpoints at 1M steps, and 2e-5 for checkpoints at 2M pre-training steps. All tests use 1000 bootstrap samples, in paired mode on the five seeds for which both 1M and 2M steps are available.

## E  CASE STUDY: MULTIBERTS VS. ORIGINAL BERT

As an additional example of application, we discuss challenges in reproducing the performance of the original BERT checkpoint, using the Multi-Bootstrap procedure.

The original `bert-base-uncased` checkpoint appears to be an outlier when viewed against the distribution of scores obtained using the MultiBERTs reproductions. Specifically, in reproducing the training recipe of Devlin et al. (2019), we found it difficult to simultaneously match performance on all tasks using a single set of hyperparameters. Devlin et al. (2019) reports training for 1M steps. However, as shown in Figure 1 and 2, models pre-trained for 1M steps matched the original checkpoint on SQuAD but lagged behind on GLUE tasks; if pre-training continues to 2M steps, GLUE performance matches the original checkpoint but SQuAD performance is significantly higher.

The above observations suggest two separate but related hypotheses (below) about the BERT pre-training procedure.

1. On most tasks, running BERT pre-training for 2M steps produces better models than 1M steps.
2. The MultiBERTs training procedure outperforms the original BERT procedure on SQuAD.

Let us use the Multi-Bootstrap to test these hypotheses.

### E.1  HOW MANY STEPS TO PRETRAIN?

Let $f$ be the predictor induced by the BERT pre-training procedure using the default 1M steps, and let $f'$ be the predictor resulting from the proposed intervention of training to 2M steps. From a glance at the histograms in Figure 8, we can see that MNLI appears to be a case where 2M is generally better, while MRPC and RTE appear less conclusive. With the MultiBERTs, we can test the significance of the results. The results are shown in Table 5. We find that MNLI conclusively performs better ($\delta = 0.007$ with $p < 0.001$) with 2M steps; for RTE and MRPC we cannot reject the null hypothesis of no difference ($p = 0.14$ and $p = 0.56$ respectively).

As an example of the utility of this procedure, Figure 7 shows the distribution of individual samples of $L$ for the intervention $f'$ and baseline $f$ from this bootstrap procedure (which we denote as $L'$ and $L$, respectively). The distributions overlap significantly, but the samples are highly correlated due to the paired sampling, and we find that individual samples of the *difference* $(L' - L)$ are nearly always positive.

### E.2  DOES THE MULTIBERTS PROCEDURE OUTPERFORM ORIGINAL BERT ON SQUAD?

To test our second hypothesis, i.e., that the MultiBERTs procedure outperforms original BERT on SQuAD, we must use the unpaired Multi-Bootstrap procedure. In particular, we are limited to the case in which we only have a point estimate of $L'(S)$, because we only have a single estimate of the performance of our baseline model $f'$ (the original BERT checkpoint). However, the Multi-Bootstrap procedure still allows us to estimate variance across our MultiBERTs seeds and across the examples in the evaluation set. On SQuAD 2.0, we find that the MultiBERTs models trained for 2M

steps outperform original BERT with a 95% confidence range of 1.9% to 2.9% and $p < 0.001$ for the null hypothesis, corroborating our intuition from Figure 2.

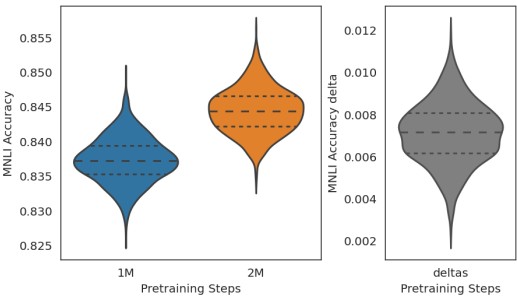

Figure 7: Distribution of estimated performance on MNLI across bootstrap samples, for runs with 1M or 2M steps. Individual samples of $L(S, (X, Y))$ and $L'(S, (X, Y))$ on the left, deltas $L'(S, (X, Y)) - L(S, (X, Y))$ shown on the right. Bootstrap experiment is run as in Table 5, which gives $\delta = 0.007$ with $p < 0.001$.

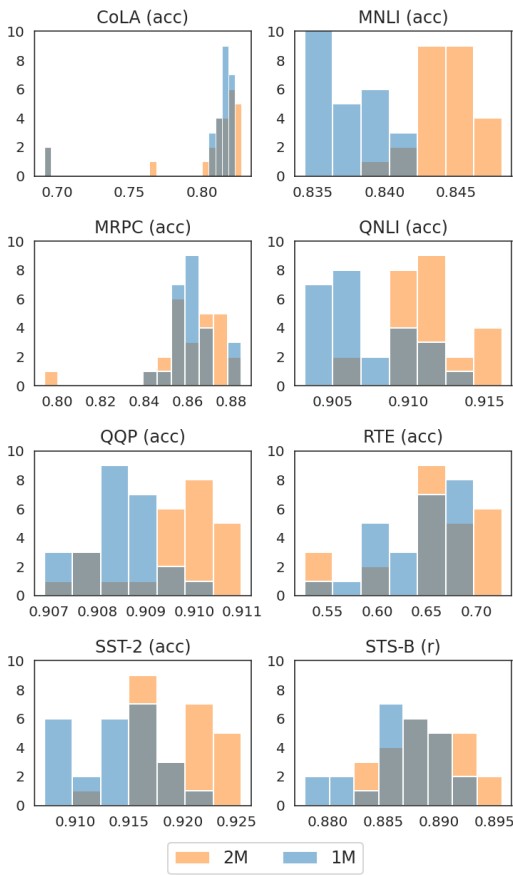

Figure 8: Distribution of the performance on GLUE dev sets, showing only runs with the best selected learning rate for each task. Each plot shows 25 points (5 fine-tuning x 5 pre-training) for each of the 1M and 2M-step versions of each of the pre-training runs for which we release intermediate checkpoints (§2).

