# OpenReview forum: "The MultiBERTs: BERT Reproductions for Robustness Analysis"
_ICLR.cc/2022/Conference — ICLR 2022 Spotlight_

### Official Review · Reviewer_H2Mp · 2021-11-02

**Correctness:** 3
**Technical Novelty And Significance:** 3
**Empirical Novelty And Significance:** 2
**Recommendation:** 6
**Confidence:** 3

**Main Review:**

The paper presents an interesting approach, a novel contribution that might have some applications. From the methodological point of view, it is well-written, there is a good review of related works, a description of the MultiBERTs release, and an application to reduce a type of gender bias in coreference systems built on BERT. There are also comprehensive supplementary materials with proofs of theorems and lemmas. It seems to be a novel approach, it's hard to assess its significance but the presented application suggests there the impact might be important. If the article is accepted, there are some typos that should be fixed before publication:
- p. 2: seeks to to analyze -> seeks to analyze
- p. 2:  the uncertainty the test sample ->  the uncertainty of the test sample
- p. 6: it looks that in one case, to mark a distribution, D, is used, in another case, it is P
- p. 8: statiatically -> statistically






**Summary Of The Paper:**

The paper presents MultiBERTs, a set of 25 model checkpoints to support robust research on BERT, and the Multi-Bootstrap, a non-parametric statistical method to estimate the uncertainty of model comparisons across multiple training seeds. It demonstrates the utility of these resources by showing how to quantify the effect of an intervention to reduce a type of gender bias in coreference systems built on BERT.

**Summary Of The Review:**

The paper presents an interesting approach, a novel contribution that might have some applications. From the methodological point of view, it is well-written, there is a good review of related works, a description of the MultiBERTs release, and an application to reduce a type of gender bias in coreference systems built on BERT. There are also comprehensive supplementary materials with proofs of theorems and lemmas. It seems to be a novel approach, it's hard to assess its significance but the presented application suggests that the impact might be important. Therefore, from the methodological point of view, the paper is marginally above the acceptance threshold, but if it is accepted, there are some minor typos that should be fixed before publication.

---

> ### Author Response · Authors · 2021-11-17
> **Reply to Reviewer H2Mp**
>
> We thank you for your review.
>
> We fixed the typos and uploaded a revised version of the paper - thank you for pointing them out.
> Regarding the significance of the work, a few notes to clarify our contribution:
> * Significance: We believe that the work is important because it lets researchers reach conclusions about BERT that would be unreachable with a single checkpoint. In particular, this allows one to draw more stable conclusions and reject spurious findings, which can be very important when dealing with high-variance metrics such as bias correlations, as in the examples in Section 4 and Appendix E.
> * Breadth: Although the scope of this paper is BERT (which itself is widely used), the Multi-Bootstrap is applicable to any experimental setup that involves randomness from pre-training, fine-tuning, and finite evaluation data (see Section 3): this encompasses a significant amount of today's ML research.
> * Novelty: To the best of our knowledge, no uncertainty quantification method has been proposed in the past to capture these dimensions (as indicated in the Related Work section).

---

### Official Review · Reviewer_67rn · 2021-11-02

**Correctness:** 4
**Technical Novelty And Significance:** 3
**Empirical Novelty And Significance:** 4
**Recommendation:** 8
**Confidence:** 4

**Main Review:**

The authors address the question of using artifactual large language models as the basis of experimentation in NLP research.  In particular, they examine the extent to which results of finetuning or transfer learning can be attributed to the use a single best checkpoint of drawn from a distribution over training data, loss, random seed, and architecture, or  for common building blocks (here they consider BERT).

Strengths:

- The biggest strength of the work as I read it is the multi-bootstrap.  This is the most likely element to be reused in future work
- I commend the openness of the authors; their code, their checkpoints, and the clarity of their writing made this paper a pleasure to review
- The inclusion of a case study in section 4 brought the whole of the contributions together, and highlighted the value of both multiple independent samples as well as the bootstrap framework used to assess the contribution of different possible choices.

Weaknesses:

- I hope that the resources of the MultiBERT models will be reused in NLP research, but fear that since LLM research is progressing at a frenetic pace, BERT is no longer as relevant as it once was even a mere 12 months ago, being supplanted on several tasks by prompt-based language models (e.g T5, GPT-3) or smaller but more versatile reimplementations (e.g GPT-J).  This is beyond the authors control however, and does not much diminish their contribution here.

- On page 5, it was not clear to me why in the paired samples design that estimating $\hat{\delta} - \delta$ represents the *overall error*.  It would help here to explain (in a footnote if need be) how the overall error is represented in terms of $\delta$ or $\hat{\delta}$.

Questions:

- This may be already established in other bootstrap results, but what is the sample efficiency of the convergence in distribution for Theorem 1?

- As the authors detail in their environmental impact statement of section 2.1, the cost of producing the MultiBERT checkpoints even in favourable energy generation conditions conditions is not negilible.  Could the multi-bootstrap be adapted to probing smaller sets of models that themselves are resampled (e.g. under repeated dropout mask as in MC-Dropout) to approximate posterior distributions over the model parameters?  The samples would no longer be IID, but such a scheme would admit many more usecases for the multi-bootstrap.

- In section 4, CDA-full sounds like a much more drastic intervention.  The MultiBERTS in CDA-incr are trained on much larger corpii, but CDA-full MultiBERTS are trained from initialization only on Webster et al’s data?  Won’t this introduce a confound whereby the CDA-full models are less proficient overall compared to the CDA-incr models?


**Summary Of The Paper:**

Many tasks in contemporary NLP begin by building off of a large language model.  This can cast the downstream task as a sort of fine-tuning experiment, whereby the results are heavily influenced by conditioning on the starting point of a single pre-trained version of an LLM.  In this work, the authors take BERT as an example, and ask how much a specific artifact as a draw from the distribution over (model weights, initialization scheme, training data, loss function) affects downstream tasks built upon it.

The authors provide a wide variety of BERT models that are varied in their training and initalization.  They define a bootstrap procedure for the scenario where multiple instantiations of base models are available, and tie their findings together in a case study of gender bias in coreference resolution.

**Summary Of The Review:**

It's a sound paper, well overdue in the NLP literature, and reminiscent of other sober reflection papers in the vein of [Melis et al. 2017, Narang et al. 2021], though the latter concentrate mainly on testing the benefit architectural improvements.  Going beyond these papers however, the authors build and provide a bootstrap procedure to evaluate specific hypotheses.

While I'm not sure that ICLR is the best venue for this work, it is clearly important and deserves to be showcased.

---

> ### Author Response · Authors · 2021-11-17
> **Reply to Reviewer 67rn**
>
> Thank you for the review and thoughtful feedback. We agree that the Multi-bootstrap method has a lot of potential for re-use in other work. We appreciate the attention to detail in the review and helpful questions that we have addressed or clarified, improving the paper overall. In particular, we clarified the sense in which $\hat{\delta} - \delta$ refers to the overall error, and have responses to the following questions below:
>
> * Sample efficiency:
> It’s typical to focus on consistency for inference results, although the bootstrap often enjoys well-quantified fast rates of convergence. There are some subtleties in the theoretical analysis due to the structure of the problem that make it difficult to analyze the rate of convergence here. We added some discussion of this along with the proofs.
>
> * Approximation:
> Thank you very much for this suggestion. We acknowledge that fine-tuning the full set of MultiBERTs checkpoint is computationally heavy, and we believe that there is opportunity here for future work. Methods such as MC-Dropout break the independence assumption on which the Multibootstrap is based, and therefore understanding the quality of the approximation, for the purpose of bootstrapping, will require further research.
>
> * Confounds in CDA:
> Thank you for pointing this out. The corpora used for pre-training are of similar size, since CDA-full's corpus was obtained by swapping pronouns in BERT's corpus. There are minor differences between the data and training hyperparameters however, we measured overall accuracy on Winogender (Section 4.1), and found no significant difference.

---

### Official Review · Reviewer_LeNr · 2021-11-03

**Correctness:** 4
**Technical Novelty And Significance:** 3
**Empirical Novelty And Significance:** 4
**Recommendation:** 8
**Confidence:** 3

**Main Review:**

Strengths:
- The paper provides various pre-trained BERT models with many checkpoints, which could enable future researches on reproducibility and robustness.
- The proposed Multi-Bootstrap procedure could give a reasonable estimate of the distribution of the error over the seeds and test instances, under both paired and unpaired scenarios.
- The provided code for Multi-Bootstrap is easy to understand.

Overall, I feel it is still expensive for common NLP tasks to adopt MultiBERTs and MultiBootStrap methods to draw conclusions, which may hinder it from a wide range of applications.

**Summary Of The Paper:**

This paper releases MultiBERTs, a set of 25 BERT base checkpoints to facilitate studies of robustness to parameter initialization and order of training examples. It also proposes the Multi-Bootstrap method to quantify the uncertainty of experimental results based on multiple pre-training seeds.

**Summary Of The Review:**

The paper provides adequate models and a good estimation method for future researches on robustness.

---

> ### Author Response · Authors · 2021-11-17
> **Reply to Reviewer LeNr**
>
> We are glad that the potential for applications on future research, the correctness and breadth of the method, and the simplicity of the code provided in the paper come through clearly. Thank you for your positive review and feedback.
>
> Regarding the cost of running experiments: we acknowledge that fine-tuning on multiple checkpoints requires additional compute, however these experiments are necessary to draw robust conclusions when metrics vary significantly across checkpoints. There are opportunities for future research on approximations, as suggested by reviewer 67rn.

---

### Official Review · Reviewer_KA6v · 2021-11-18

**Correctness:** 4
**Technical Novelty And Significance:** 3
**Empirical Novelty And Significance:** 4
**Recommendation:** 6
**Confidence:** 2

**Main Review:**

The paper presents MultiBERTs, a set of 25 model checkpoints, and the Multi-Bootstrap, a non-parametric method to estimate the model uncertainty. The experiments verified the proposed method on a case study of gender bias in coreference resolution.

Paper Strengths:
- It is great to see the models and statistical library are available online (165 checkpoints). I appreciate the authors can provide the CO2 information on model training, which is environment friendly.
- This is an empirical analysis paper, which is useful for the community but the computational cost seems very high. Pretrained model has been widely used and show impressive performance. The method in this paper is novel and theoretically sound, which is helpful for understanding the large models.
- The proposed method is rigorous in terms of mathematics and statistics. Even though the techniques are simple, but they can be categorized into three designs and formulated as a formal problem.


**Summary Of The Paper:**

The paper presents MultiBERTs, a set of 25 model checkpoints, and the Multi-Bootstrap, a non-parametric method to estimate the model uncertainty. The experiments verified the proposed method on a case study of gender bias in coreference resolution.

**Summary Of The Review:**

This paper is different from classical analysis paper in NLP. The authors claim to make progress on language model pre-training and it is essential to distinguish between the properties of specific model artifacts and those of the training procedures that generated them. I can buy this claim to some degree but still reserve my recommendation for acceptance.

---

> ### Author Response · Authors · 2021-11-19
> **Reply to Reviewer KA6v**
>
> We thank you for your review. We are glad that you find our method theoretically sound and its implementation useful for the community.
>
> We would like to clarify that this is not a classical analysis paper; our main contributions are a resource (the checkpoints) and a method (the Multi-Bootstrap) that can be used by other ML researchers across the community. The analysis in Section 4 serves to demonstrate the practical importance of these methodological contributions by (1) illustrating that one may reach different conclusions when studying a general model rather than a specific checkpoint, and (2) demonstrating how to use the resources that we introduce in this context.

---

### Decision · Program_Chairs · 2022-01-20

**Decision:**

Accept (Spotlight)

**Comment:**

This paper is a resource and numerical investigation into the variability of BERT checkpoints. It also provides a bootstrap method for making investigations on the checkpoints.

All reviewers appreciate this contribution that can be expected to be used by the NLP community.